



# Airborne particles in the Brazilian city of São Paulo: One-year investigation for the chemical composition and source apportionment

Guilherme Martins Pereira [1,4], Kimmo Teinilä [2], Danilo Custódio [1,3], Aldenor Gomes Santos [4], Huang Xian [5], Risto Hillamo [2], Célia A. Alves [3], Jailson Bittencourt de Andrade[4,6,7], Gisele Olímpio da Rocha[4,6,7], Prashant Kumar[8,9], Rajasekhar Balasubramanian [5], Maria de Fátima Andrade [10], Pérola de Castro Vasconcellos[1,4]

[1]*Institute of Chemistry, University of São Paulo, São Paulo – SP, 05508-000, Brazil.*
[2]*Finnish Meteorological Institute, P.O. Box 503, FI-00101 Helsinki, Finland.*
[3]*CESAM & Department of Environment, University of Aveiro, Aveiro, 3810-193, Portugal*
[4]*INCT for Energy and Environment, Federal University of Bahia, Salvador–BA, 40170-115, Brazil*
[5]*Department of Civil and Environmental Engineering, National University of Singapore, E1A 07-03, 117576, Singapore*
[6]*CIEnAm, Federal University of Bahia, Salvador–BA, 40170-115, Brazil*
[7]*Institute of Chemistry, Federal University of Bahia, Salvador–BA, 40170-115, Brazil*
[8]*Department of Civil and Environmental Engineering, Faculty of Engineering and Physical Sciences, University of Surrey, Guildford GU2 7XH, United Kingdom*
[9]*Environmental Flow Research Centre, Faculty of Engineering and Physical Sciences, University of Surrey, Guildford GU2 7XH, United Kingdom*
[10]*Institute of Astronomy, Geophysics and Atmospheric Sciences, University of São Paulo, São Paulo–SP, 05508-090, Brazil*

*Correspondence to: Guilherme Martins Pereira (martinspereira2@hotmail.com)*

**Abstract**. São Paulo in Brazil has relatively relaxed regulations for ambient air pollution standards and often experiences high air pollution levels due to emissions of airborne particles from local sources and long-range transport of biomass burning-impacted air masses. In order to evaluate the sources of particulate air pollution (PM) and related health risks, a year-round sampling was performed for $PM_{2.5}$ ($\leq 2.5$ µm) and $PM_{10}$ ($\leq 10$ µm) in 2014 through intensive (every day sampling in wintertime) and extensive campaigns (once a week for the whole year) with 24 h of sampling. This year was characterized to have lower average precipitation comparing to meteorological data, and high pollution episodes were observed all year round, with a significant increase of pollution level in the intensive campaign, which was performed during wintertime. Different chemical constituents, such as carbonaceous species, polycyclic aromatic hydrocarbons (PAHs) and derivatives, water-soluble ions and biomass burning tracers were identified in order to evaluate health risks and to apportion sources. The species such as PAHs, inorganic and organic ions and monosaccharides were determined by chromatographic techniques and carbonaceous species by thermal-optical analysis. The associated risks to particulate matter exposure based on PAH concentrations were also assessed, along with indexes such as the benzo[a]pyrene equivalent (BaPE) and lung cancer risk (LCR). High BaPE and LCR were observed in most of the samples, rising to critical values in the wintertime. Also, biomass burning tracers and PAHs were higher in this season, while secondarily formed ions presented low variation throughout the year. Meanwhile, vehicular tracer species were also higher in the intensive campaign suggesting the influence of lower dispersion conditions in that period. Source apportionment was done by Positive Matrix Factorization (PMF), which indicated five different factors: road dust, industrial emissions, vehicular exhaust, biomass burning and secondary processes. The results highlighted the contribution of vehicular emissions and the significant input from biomass combustion in wintertime, suggesting that most of the particulate matter is due to local sources, besides the influence of pre-harvest sugarcane burning.





## 1    Introduction

Air pollution caused by atmospheric particulate matter (PM) is one of the major environmental problems encountered in Latin American cities such as São Paulo (Brazil), Mexico City (Mexico), Bogota (Colombia) and Santiago (Chile) (Romero-Lankao et al., 2013; Vasconcellos et al., 2010, 2011a; Villalobos et al., 2015). The air pollution thresholds in most of the Latin American cities are not very stringent compared to international standards or guidelines (Alvarez et al., 2013; Kumar et al., 2016). Several studies have highlighted a statistical relation between PM and health problems, including respiratory and

cardiovascular diseases and genotoxic risks (Newby et al., 2015; de Oliveira Alves et al., 2014; Pope, 2000).  In this context, $PM_{2.5}$ (PM with aerodynamic diameter smaller than 2.5 µm) and $PM_{10}$ (PM with aerodynamic diameter smaller than 10 µm) are particles that are able to penetrate in the respiratory system, with $PM_{2.5}$ reaching alveoli in the lungs, and induce adverse impacts on human health (Cai et al., 2015; Kumar et al., 2014). The elderly and the children are more prone to be susceptible individuals to the health effects resulting from $PM_{2.5}$ (Cançado et al., 2006; Segalin et al., 2017). Considering that elderly

population has grown in São Paulo over the last decades (SEADE, 2016; Segalin et al., 2017), the PM health-related issues can become more relevant. PM also plays an important role in ecosystem biogeochemistry, hydrological cycle, cloud formation and atmospheric circulation (Pöschl, 2005).

Carbonaceous species as organic and elemental carbons (OC and EC) represent a large fraction of PM and play an important role in the formation of haze, interaction with climate and adverse human health effects (Bisht et al., 2015; Liu et al., 2016;

Seinfeld and Pandis, 2006). Water-soluble ions (WSI) account for another major fraction of aerosols in urban areas and are able to affect visibility, particle hygroscopicity, cloud formation; they also influence acidity in rainwater and impact climate (Cheng et al., 2011; Jung et al., 2009; Khoder and Hassan, 2008; Tan et al., 2009; Tang et al., 2016; Yang et al., 2015).

Particulate organic carbon includes key species including polycyclic aromatic hydrocarbons (PAHs) and monosaccharides are considered as biomass burning tracers (such as levoglucosan, mannosan, and galactosan) (Simoneit et al., 1999). PAHs

have natural sources, but are mostly formed by anthropogenic emissions. They have been studied because of their carcinogenic properties (de Oliveira Alves et al., 2014; Seinfeld and Pandis, 2006). The nitrated and oxygenated PAHs (nitro- and oxy-PAHs) are emitted as primary species or are formed in situ as secondary compounds (Kojima et al., 2010; Souza et al., 2014b; Zhou and Wenger, 2013; Zimmermann et al., 2013). They are potentially more mutagenic and/or carcinogenic than their PAH precursors (Franco et al., 2010).

Chemical speciation and PAH risk assessment have been performed in several Latin American sites, specifically in urban São Paulo, Bogota, Buenos Aires (Vasconcellos et al., 2011a; Vasconcellos et al., 2011b) and in forested areas such as the Amazon region (de Oliveira Alves et al., 2015). Biomass burning tracers have been detected in high concentrations in São Paulo during the dry season and are attributed to the long-range transport of aerosols from areas affected by sugarcane burning. Source apportionment studies have been carried out in São Paulo (Table 1) in the last three decades, but not in as

much detail as in other megacities. Detailed characterization of the organic fraction of aerosols is still scarce.

A previous study performed in São Paulo in 1989 highlights the relative importance of the emissions from residual oil and diesel in $PM_{2.5}$ and soil dust in the coarse grain size (Andrade et al., 1994). Da Rocha et al. (2012) studied the emission sources of fuel and biomass burning, the gas-to-particle conversion, and sea spray emissions in PM in São Paulo, in one year period (between 2003 and 2004). Another study conducted in the winter of 2003 pointed out a strong impact of local sources

in three sites in the state of São Paulo, besides the influence of remote sources (Vasconcellos et al., 2007). A source apportionment for PAHs in the winter of 2002 reported a predominance of diesel emissions for the polyaromatics in $PM_{2.5}$ (Bourotte et al., 2005). In turn,  Castanho and Artaxo (2001), in their study of 1997 and 1998 in São Paulo city, reported no significant differences in the main air pollution sources (i.e. automobile traffic and soil dust) between wintertime and summertime. The main sources for $PM_{2.5}$ were automobile traffic and soil dust. However, biomass burning was not

considered as a potential source by the authors.





The current study presents a more comprehensive study that should lead to a better understanding of the main PM sources and atmospheric processes occurring in the São Paulo megacity than previous studies reported in the literature. A year of extensive sampling of aerosol ($PM_{2.5}$ and $PM_{10}$) and a wintertime intensive campaign were performed. Different classes of chemical components in PM were determined such as carbonaceous species, WSI, monosaccharides, PAHs, and their

derivatives. Meteorological data were also collected during the sampling days. Moreover, the benzo[a]pyrene equivalent (BaPE) and lung cancer risk (LCR) indexes were calculated in order to assess the potential toxicity of PAHs. Positive Matrix Factorization (PMF) analysis was also used for the source apportionment of $PM_{10}$ during the extensive campaign.

**Methodology**

**1.1  Sampling campaigns**

Aerosol samples were collected at a São Paulo site (SPA, 23°33′34″S and 46°44′01″W) located on the rooftop of the

Atmospheric Sciences Department, at Institute of Astronomy and Atmospheric Sciences (IAG-USP) building, within the campus of University of São Paulo. The location is inside a green area and approximately 2 km away from an important expressway (Marginal Pinheiros) (Fig. 1). Aerosols were collected in intensive (every day) and extensive campaigns (once a week) throughout 2014. Firstly, the extensive campaign was performed weekly. Accordingly, samples were collected every

Tuesday for $PM_{2.5}$ (termed $Ext_{2.5}$ in this study) and $PM_{10}$ (termed $Ext_{10}$). However, due to equipment breaking down, the $PM_{2.5}$ sampling was stopped in September while the $PM_{10}$ sampling continued until December ($n = 32$ and $38$, respectively). Secondly, the intensive campaign (termed $Int_{2.5}$) took place between 01 and 18 July, 2014 ($n = 12$), only for $PM_{2.5}$ due to problems with $PM_{10}$ equipment. However, there were four days (between 08 and 11 July) for which data were not collected due to heavy rain.

PM samples were collected for a period of 24 h, with high-volume air samplers (Hi-Vol), with a flow rate of 1.13 $m^3$ $min^{-1}$, with 2.5 and 10 µm size selective inlets (Thermo Andersen, USA). Prior to sampling, quartz fiber filters (20 cm × 25 cm, Millipore, USA) were baked for 8 h at 800 °C to remove the organics. In addition, filters were equilibrated at room temperature and weighed in a microbalance before and after the sampling, in order to estimate the PM concentration. After sampling and weighing, the filters were wrapped in aluminum foil and stored in a refrigerator at 5 °C until chemical analyzes

were performed.

**1.2  Meteorological data and gaseous species**

The meteorological data (ambient temperature, relative humidity, precipitation and wind speed) were collected from the climatological bulletin of IAG/USP meteorological station (IAG, 2014). The climate of São Paulo is often classified as humid subtropical (Andrade et al., 2012a). The wintertime in the city is characterized by a slight decrease in temperatures,

together with considerably lower relative humidity and precipitations, with more thermodynamic stability, often resulting in accumulation of air pollutants in the lower troposphere, being also subjected to thermal inversion episodes (Miranda et al., 2012). The local air circulation is mainly associated with the Atlantic Ocean breeze and cold fronts in wintertime often intensified that, with winds generally coming from Southeast (Vasconcellos et al., 2003). In Fig. S1 is presented the comparison between the average climatological temperature and the data for 2014 (IAG, 2014). During the 2014 campaign,

the summer was atypically warmer and dry.

In order to analyze the long-range transport of air pollutants, backward air mass trajectories (96 h) were run using the HYSPLIT model (Draxler and Rolph, 2003), through READY (Real-time Environmental Applications and Display System) platform from NOAA (National Oceanic and Atmospheric Administration). The considered heights were 500, 1500 and



3000 m, corresponding to trajectories near the ground, upper boundary layer and low free troposphere, respectively (Cabello et al., 2016; Toledano et al., 2009).

### 1.3   Analytical procedures, reagents and standards

After sampling, the filters were punched for the chemical analysis, as shown in Table 2, which lists all substances determined and their respective analytical techniques as well as their detection limits (DL).

Carbonaceous species were determined at the University of Aveiro, with two punches of 9 mm diameter. Firstly, the

carbonates were removed with hydrochloric acid fumes and then OC and EC were determined by a thermal-optical transmission equipment developed at the university. The system comprises a quartz tube with two heating zones, a pulsed laser and a non-dispersive infrared $CO_2$ analyzer (NDIR). The filters were placed into the first heating zone of the quartz tube then heated to 600 $^o$C in a nitrogen atmosphere for the organic fraction to vaporize, which was quantified as OC. EC was determined with a sequential heating at 850 $^o$C in an atmosphere containing 4 % $O_2$. The other heating zone was filled

with cupric oxide and was maintained at 650 $^o$C in a 4 % $O_2$ atmosphere, to assure that all carbon is volatilized to $CO_2$, which is quantified by a NDIR analyzer (Alves et al., 2015).

The determination of polycyclic aromatic hydrocarbons and their derivatives was done at Federal University of Bahia, Brazil, and is summarized in Table 2. Briefly, samples were extracted for 23 min in an ultrasonic bath (4.2 cm$^2$ punches) with a 500 μL solution of 18 % of acetonitrile in dichloromethane, employing miniaturized extraction devices (Whatmann

Mini$^{TM}$ UniPrep Filters, Whatman, USA). Their quantification was carried out by gas chromatography with high-resolution mass spectrometer detection (GC-MS). The procedure is described in more details in Santos et al. (2016). BeP was quantified with the same calibration curve as BaP since they have similar fragmentation pattern in the MS detector (Robbat and Wilton, 2014).

US Environmental Protection Agency (EPA) 610 PAH mix in methanol:dichloromethane (1:1), containing 2000 μg mL$^{-1}$

each, was purchased from Supelco (St. Louis, USA). Individual standards of 50 μg mL$^{-1}$ coronene (Cor) and 1000 μg mL$^{-1}$ perylene (Per) and two deuterated compounds, pyrene D10 (Pyr d10) and fluorene D10 (Flu d10) were purchased from Sigma-Aldrich (St. Louis, USA. Quinones investigated in this study were purchased from Fluka (St. Louis, USA). Nitro-PAH certified standard solutions SRM 2264 (aromatic hydrocarbons nitrated in methylene chloride I) and SRM 2265 (polycyclic aromatic hydrocarbons nitrated in methylene chloride II) were purchased from the National Institute of Standards

and Technology (NIST, USA).

Monosaccharides and WSI were determined at Finnish Meteorological Institute. From quartz fiber filters samples, 1 cm$^2$ filter pieces were punched for both analyzes. Concentrations of monosaccharides were determined using a Dionex ICS-3000 system coupled to a quadrupole mass spectrometer (Dionex MSQ™) by high-performance anion exchange chromatography (HPAEC-MS). Levoglucosan (1,6-anhydro-β-D-glucopyranose, purity 99 %; Acros Organics, NJ, USA), mannosan (1,6-

anhydro-β-D-mannopyranose; purity 99 %; Sigma-Aldrich Co., MO, USA), and galactosan (1,6-anhydro-β-D-galactopyranose; Sigma-Aldrich Co.) were used for the calibration. The 1 cm$^2$ punches were extracted with 5 mL of deionized water (Milli-Q water; resistivity 18.2 MΩ.cm at 25 ∘C, Merck Millipore, MA, USA) with methyl-β-D-arabinopyranoside as internal standard (purity 99 %; Aldrich Chemical Co., WI, USA), and 10 min of gentle rotation. The extract was filtered through an IC Acrodisc® syringe filter (13 mm, 0.45 μm Supor® (PES) membrane, Pall Sciences)

(Saarnio et al., 2010).

In order to determine the water-soluble ions ($Cl^-$, $NO_3^-$, $SO_4^{2-}$, $C_2O_4^{2-}$, methylsulfonate, $Na^+$, $K^+$, $NH_4^+$) 10 mL of deionized water was used to extract the sample aliquots, with 10 min of gentle rotation. The ions were determined using two ion chromatography systems (ICS 2000 system, Dionex) simultaneously; cations were analyzed using a CG12A/CS12A column with an electrochemical suppressor (CSRS ULTRA II, 4 mm) and anions using an AG11/AS11 column with an

electrochemical suppressor (ASRS ULTRA II, 4 mm).





Finally, trace elements in the samples were extracted using a microwave digestion system (MLS-1200 mega, Milestone Inc., Italy) at National University of Singapore. Punches of the filters were cut into small pieces and added into PTFE vessels with 4 mL $HNO_3$ (Merck), 2 mL $H_2O_2$ (Merck) and 0.2 mL HF (Merck). The vessels were then subjected to a three-stage digestion inside the microwave digester (250 W for 5 min, 400 W for 5 min, and 600 W for 2 min). Following the digestion

procedure, extracts were filtered with 0.45 µm PTFE syringe filters, diluted 8 times and stored in the 4 ˚C cold room. The concentrations of trace elements were quantified by ICP-MS (Agilent 7700, USA) in triplicates. The instrumental parameters maintained during sample runs using the ICP-MS analysis were: plasma gas (15.0 L $min^{-1}$), auxiliary gas (1.0 L $min^{-1}$), and nebulizer gas (1.0 L $min^{-1}$). Clean ceramic scissors and forceps were used to handle all PM samples. ICP-MS standards (purchased from High-Purity Standards, USA) were used for calibration.

**1.4   Statistical analysis and receptor model**

Pearson coefficients were calculated to verify the correlation between all the species (software STATISTICA). It determines the extent to which values of the variables are linearly correlated. The coefficients ($r$) were considered significant when $p <$ 0.05. Two-tailed t-tests were also employed in order to evaluate equal and unequal variances ($p < 0.05$). Polar plots considered the mass concentrations as a function of wind speed and direction (software R x64 3.3.2).

The widely used source apportionment model, positive matrix factorization (PMF), was applied to the $PM_{10}$ dataset (Paatero and Tapper, 1994). In this study, specifically, the EPA PMF5.0 software was used. Variables were classified as *strong*, *weak* and *bad* according to the signal-to-noise ratio (*S/N*), number of samples below the detection limit (Amato et al., 2016; Contini et al., 2016; Paatero and Hopke, 2003) and thermal stability of the species. The species were categorized as *bad* when the S/N ratios were less than 0.2 and *weak* when  the S/N ratios were greater than 0.2 but less than 2 (Lang et al.,

2015). Accordingly, species with S/N ratios higher than 2 were considered *strong*. *Bad* variables were excluded from the model and the *weak* ones had their uncertainty increased by a factor of 3, as described in the EPA PMF Fundamentals and User Guide (Norris et al., 2014).

When concentrations were below the detection limits, they were substituted by half the detection limit (*DL)*. Missing data were replaced by the median (*M*) of the whole dataset for that species. Uncertainties were calculated by Eq. (1) according to

Norris et al. (2014), when the concentrations were below the detection limits:

$$Unc = 5/6 \times DL \tag{1}$$

Uncertainty for missing data (Brown et al., 2015)  is given by Eq. (2):

$$Unc = 4 \times M \tag{2}$$

When the concentrations were above the detection limit, uncertainty is determined from Eq. (3):

$$Unc = ([EF \times EC]^2 + [0.5 \times DL]^2)^{1/2} \tag{3}$$

Where *EF* is the error fractions and *EC* is the element concentration.

Q robust value ($Q_R$) is the goodness-of-fit parameter computed with the exclusion of points not fitted by the model. To evaluate the number of factors, $Q_R$ was compared to $Q_T$ (Q theoretical value). At the point when changes in the ratio $Q_R/Q_T$ become smaller with the increase of the number of factors, it can be demonstrative that there might be an excessive number

of factors being fitted (Brown et al., 2015). $Q_T$ was estimated as in Lang et al. (2015), given by Eq. (4):

$$Q_T = (n_s \times n_e) - ([n_s \times n_f] + [n_e \times n_f]) \tag{4}$$

Where $n_s$ is the number of samples, $n_e$ is the number of *strong* elements, and $n_f$ is the number of factors.





## 2    Results and discussions

### 3.1    Concentrations of $PM_{2.5}$ and $PM_{10}$ during extensive campaigns

The extensive campaigns ($Ext_{2.5}$ and $Ext_{10}$) were carried out over a whole year, during which the meteorological conditions varied largely. The average temperature during the sampling days in all campaigns ranged from 14 to 26 °C and the wind speed varied between 0.6 and 2.6 m s$^{-1}$; most of the sampling was carried out on days without rainfall. In Fig. 2 is presented the meteorological variables, $PM_{10}$ and $PM_{2.5}$ concentration for all analyzed days.

There was a moderate negative correlation between $PM_{10}$, $PM_{2.5}$ and minimum relative humidity and average wind speed (Table S1). This observation is in agreement with the fact that days with lower relative humidity and lower wind speed present higher $PM_{2.5}$ and $PM_{10}$ levels than in more humid and with more ventilation conditions.

In the extensive campaign, the PM mass concentrations exhibited a wide range of concentrations. For example, $Ext_{2.5}$ ranged from 8 to 78 µg m$^{-3}$ (average 30 µg m$^{-3}$), whereas $Ext_{10}$ values varied between 12 and 113 µg m$^{-3}$ (average 44 µg m$^{-3}$) (Fig. 3). The World Health Organization (WHO) recommends a daily limit for $PM_{10}$ of 50 µg m$^{-3}$ and of 25 µg m$^{-3}$ for $PM_{2.5}$, (WHO, 2006) while the Brazilian Environmental Agency (CONAMA) recommends a threshold of 150 µg m$^{-3}$ for $PM_{10}$ (CONAMA, 1990; Pacheco et al., 2017). In our study, 50 % of the $Ext_{2.5}$ and 30 % of the $Ext_{10}$ samples were above the guidelines recommended by WHO. When considering the CONAMA standards, only one day in the extensive campaign was near the target limit. The $Ext_{10}$ campaign was divided into two periods: dry (April to September) and rainy (October to March). It was observed that the average $PM_{10}$ was 52 µg m$^{-3}$ in the dry period (above the guideline recommended by WHO) and of 35 µg m$^{-3}$ in the rainy period (below the same guideline).

A study done by Vasconcellos et al. (2011b) about a decade ago (2003/2004) in the city, showed a similar average of $PM_{10}$ (46 µg m$^{-3}$). According to CETESB (São Paulo State Environmental Agency), the annual average $PM_{10}$ concentrations (considering all monitoring stations in the São Paulo Metropolitan Area) ranged from 33 to 41 µg m$^{-3}$, between the years of 2005 and 2014, showing no significant differences (CETESB, 2015).

The average values for $PM_{2.5}$ were higher than those obtained in a year study done in traffic sites in two European cities: London and Madrid in 2005 (warm period: 19.40 and 20.63 µg m$^{-3}$ for $PM_{2.5}$, respectively) (Kassomenos et al., 2014). The European Union has a more restrictive control of pollutant emissions since an annual mean of 40 µg m$^{-3}$ is established for $PM_{10}$ and a limit value of 25 µg m$^{-3}$ is imposed for $PM_{2.5}$ (Kassomenos et al., 2014). However, these averages in São Paulo are lower than in some year-round studies performed at sites from Chinese megacities, such as Shanghai (83 µg m$^{-3}$ for $PM_{2.5}$ and 123 µg m$^{-3}$ for $PM_{10}$) and Nanjing (222 µg m$^{-3}$ for $PM_{2.5}$ and 316 µg m$^{-3}$ for $PM_{10}$) (Shi et al., 2015; Wang et al., 2003, 2013a). Indeed, in 2014, Zheng et al. (2016) assessed the $PM_{2.5}$ concentrations in 161 Chinese cities reporting an annual average concentration of 62 µg m$^{-3}$.

In this study was found that, on average, more than 60 % of the total mass PM is within the $PM_{2.5}$; it is consistent with a previous study done at this site (dry season, 2008) when this value was 69 % (Souza et al., 2014a). On the other hand, in a two-year study conducted in 10 urban sites in Rio de Janeiro, the coarse fraction represented from 60 to 70 % of the $PM_{10}$ mass concentration (Godoy et al., 2009). The $PM_{2.5}/PM_{10}$ ratio found in other urban Brazilian sites with different characteristics (biomass burning, coastal environment) were close to 40 %, considerably lower than in São Paulo Metropolitan Area, according to the local environmental agency (CETESB, 2015), highlighting the importance of fine particulate matter over São Paulo city aerosol.

### 3.2    Concentrations of $PM_{2.5}$ during intensive campaign

The winter campaign began with high $PM_{2.5}$ concentrations (a maximum of 88 µg m$^{-3}$ on 02 July) and low relative humidity (minimum of 21 %). The average temperatures ranged from 15 to 21 °C and the wind speed ranged between 0.6 and 2.2 km h$^{-1}$. The concentrations of $PM_{2.5}$ in the intensive campaign ranged from 15 to 88 µg m$^{-3}$ (average 45 µg m$^{-3}$), with a similar





average to that obtained in another intensive study in 2008, 47 µg m$^{-3}$ (Souza et al., 2014a). The levels of PM$_{2.5}$ in this campaign were above the recommended by WHO in 90 % of the sampling days.

The average concentration of PM$_{2.5}$ was higher in Int$_{2.5}$ than in Ext$_{2.5}$, which can be explained by the fact that the campaign took place in the dry season (winter). In winter, the meteorological conditions are more unfavorable to the dispersion of pollutants and also due to the predominance of sugarcane burning (da Rocha et al., 2005, 2012; Sánchez-Ccoyllo and

Andrade, 2002; Vasconcellos et al., 2010).

### 3.3 WSI and trace elements

The WSI represent a large fraction in the aerosol mass and have already been suggested to present ability to form CCN (cloud condensation nuclei) and fog (Rastogi et al., 2014). The secondary inorganic components, sulfate, nitrate and ammonium (SNA) were the most abundant ions in all campaigns (Table 3), which has already been observed in previous

studies for this site (Vasconcellos et al., 2011a). SNA accounted for 74, 82 and 79 % of the total mass of inorganic species in the Int$_{2.5}$, Ext$_{2.5}$ and Ext$_{10}$ campaigns, respectively. The SNA were also found to be the major portion of the WSI in other studies around the world. For instance, Zheng et al. (2016) assessed PM$_{2.5}$ concentrations at 17 diversified sites in China. An average contribution of SNA of more than 90 % of total ions was obtained, which represented 50 % of PM$_{2.5}$. The levels of SNA in aerosols from urban sites are highly influenced by the anthropogenic emissions of precursors (SO$_2$, NO$_x$, and NH$_3$)

(Wang et al., 2013b), although they may also be directly emitted for different sources, such as automobile or industrial sources.

Sulfate average concentrations appear to vary less than nitrate comparing Int$_{2.5}$ to Ext$_{2.5}$ and were not statistically different ($p$ ~0.8). The same trend in sulfate in this study, was also observed by Villalobos et al. (2015) for Santiago, Chile, 2013. In that study, the annual average concentration of sulfate (2000 ng m$^{-3}$) is considerably lower than that observed in São Paulo

extensive campaigns in 2014. The sulfate concentrations in Santiago aerosols have reduced since air quality regulations limited to 15 ppm the sulfur content in diesel and gasoline (MMA, 2014). In Brazil, since 2013 the S-10 diesel (10 ppm of Sulphur) substituted the S-50 diesel (50 ppm), whereas in 2014 the S-50 gasoline replaced the S-800 gasoline (800 ppm), although older vehicles are still allowed to use S-500 (500 ppm) diesel (CETESB, 2015). During the studies done in several urban sites in China, sulfate concentrations varied between 4200 and 23000 ng m$^{-3}$. These values are higher than those of this

study and also 5 to 10 times higher than the measured concentrations in Europe and United States (Hidy, 2009; Putaud et al., 2004; Zheng et al., 2016).

The SO$_4^{2-}$/NO$_3^-$ ratio was nearly twice higher in the Ext$_{2.5}$ campaign than in Int$_{2.5}$. It has already been observed in warmer ambient conditions the fine NO$_3^-$ aerosols can be volatilized, increasing the ratio between these species (Rastogi and Sarin, 2009; Souza et al., 2014a). NH$_4$NO$_3$ exists in a reversible equilibrium between HNO$_3$ and NH$_3$ (Tang et al., 2016).

Ammonium concentrations were not significantly different ($p$ ~ 0.3) and were slightly higher in Int$_{2.5}$ (1712 ng m$^{-3}$) than in Ext$_{2.5}$ (1370 ng m$^{-3}$). Ratio ∑cations/∑anions (ion balance) was calculated similarly to what was done in the previous study by da Rocha et al. (2012). In the present study, the ratios were lower than 1.0 in all the campaigns, and considerably higher in Int$_{2.5}$. It is suggested that it occurred due to the lack of other cationic species data.

Non-sea-salt potassium was calculated (nss-K$^+$) based on seawater ion ratios [ss-K$^+$] = 0.036 [Na$^+$]  (Nayebare et al., 2016;

Seinfeld and Pandis, 2006). Concentrations of nss-K$^+$ were significantly higher in Int$_{2.5}$ than in Ext$_{2.5}$, with average concentrations of 809 and 366 ng m$^{-3}$, respectively ($p < 0.01$). The higher concentrations found in the intensive campaigns have already been attributed to biomass burning in previous studies (Pereira et al., 2017; Vasconcellos et al., 2011a). However, potassium ion can also come from soil resuspension (Ram et al., 2010; Tiwari et al., 2016), which becomes important in PM$_{2.5-10}$. Higher concentrations of chloride in fine particles (964 and 330 ng m$^{-3}$ for Int$_{2.5}$ and Ext$_{2.5}$,

respectively) were observed in campaign Int$_{2.5}$ (although the value of $p$ was slightly above 0.05), probably due to a higher influence of biomass burning (Allen et al., 2004). On the other hand, chloride in coarse particles is mostly attributed to





marine aerosols. $Cl^-/Na^+$ ratios were below 1.8 in $Ext_{2.5}$ and $Ext_{10}$ and higher in $Int_{2.5}$; although $Cl^-/Na^+$ ratios are attributed to increased sea salt contribution (Souza et al., 2014a), the higher contribution in intensive campaign may be explained by a higher contribution of other sources of chloride in that period, such as biomass burning.

Pearson correlations were obtained for all determined species in $Ext_{2.5}$ and $Ext_{2.5-10}$ (coarse mode), including meteorological data such as temperature, relative humidity and wind speed, some gaseous species such $NO_x$ and CO, were obtained from CETESB database and were also included. $NH_4^+$ was from moderately to strongly correlated with $C_2O_4^{2-}$ (oxalate), $Cl^-$, $NO_3^-$ and $SO_4^{2-}$ in $Ext_{2.5}$ (R = 0.66, 0.62, 0.85 and 0.79, respectively) suggesting the neutralization of oxalic, hydrochloric, nitric and sulfuric acids by $NH_3$ (Table S2). The formation of $(NH_4)_2SO_4$, a non-volatile species, could represent a gas-to-particle

process and can account for the formation of new particles through nucleation (Mkoma et al., 2014; da Rocha et al., 2005) and can lead to CCN (cloud condensation nuclei) formation. $NH_4NO_3$ and $NH_4Cl$ also have an important influence on Earth's acid deposition (Tang et al., 2016).

   $Na^+$ was strongly correlated with $Cl^-$ in $Ext_{2.5}$ (R = 0.78) and had relatively higher correlations with this species (R = 0.35) in the coarse fraction. These species are often associated with marine aerosol, which is mainly in the coarse mode (Godoy et al.,

2009; da Rocha et al., 2012). Although it was observed that ocean influence is not the only source of $Na^+$ in the site, this species may have vehicular sources (Vieira-Filho et al., 2016). $C_2O_4^{2-}$ was also moderately correlated with $NO_3^-$, $SO_4^{2-}$ and $K^+$ (R = 0.67, 0.61 and 0.68, respectively), this species reported sources can be biomass burning and secondary conversion of natural and anthropogenic gases (Custódio et al., 2016). The secondarily formed species were negatively correlated with wind speed (from R = -0.40 to R = -0.70); lower wind speed can increase the formation of secondary ionic species due to an

increase the precursor species concentrations (Yu et al., 2016).

   Average, maximum and minimum trace element concentrations are presented in Table 4. Mg, Al, K, Ca, Fe, Cu and Zn were the most abundant elements in all campaigns, similar to those observed by Vasconcellos et al. (2011a) for the intensive campaign in 2008. All of them had higher concentrations in $Int_{2.5}$ than in $Ext_{2.5}$. However, crustal elements were significantly higher; Al and Ca had concentrations nearly 3 times higher in intensive campaign ($p < 0.05$). A similar trend was observed

between wintertime and summertime campaigns by Castanho and Artaxo (2001). They reported higher concentrations of soil resuspension elements during wintertime. An increase in soil resuspension is expected in drier conditions.

   As observed for $nss-K^+$, elemental K average concentration was more than twice higher in $IC_{2.5}$ than $EC_{2.5}$ ($p < 0.05$). This may be explained by a higher biomass burning contribution during the intensive campaign since sugar cane burning is significantly increased in this time of the year. Cu has been attributed to vehicular emissions in São Paulo (Castanho and

Artaxo, 2001), because it may be present in the ethanol, which is mixed with gasoline and used in light-duty vehicles in Brazil. Cu has also been related to wear emissions of road traffic (Pio et al., 2013). This element was approximately 70 % higher in $Int_{2.5}$ than $Ext_{2.5}$. Although there is no significant difference in vehicular emissions all year round, the meteorological conditions are more unfavorable to pollutant dispersion in winter season.

   Enrichment factor (EF) is an approximation often used in order to identify the degree to which an element in an aerosol is

enriched or depleted regarding a specific source. EFs are calculated based on a reference metal (Al as a soil tracer in this study), considering crustal element composition (Lee, 1999). A convention often adopted is considering that when elements have EFs below 10 they have significant crustal source and are often called non-enriched elements (NEEs) and when the elements have EFs above 10 they have a higher non-crustal character and are referred as anomalously enriched elements (AEEs) (Pereira et al., 2007). Values were higher than 10 for Cr (except for $Int_{2.5}$), Cu, Zn, As, Se, Cd, Sn, Tl, Pb and Bi,

meaning that they can be attributed to anthropogenic sources as vehicular and industries emissions (Table S3). Elements like K, Mn, Ni, Rb, Sr, Cs, Li, Mg, Ca, Fe, Co and Sr had EFs lower than 10 and could be attributed to soil resuspension (da Rocha et al., 2012).

   Strong correlations were observed between Al and Li, Mg, K, Ca, Mn, Fe, Rb and Sr (R > 0.85) in $Ext_{2.5}$. Al also had strong correlations with Li, Mg, K, Ca and Fe in $Ext_{2.5-10}$ (R > 0.70). Strong correlations were observed between species like $Cl^-$





and $NO_3^-$ with Mg, Al, Ca and Fe (R > 0.7); atmospheric reactions can occur between acids (HCl and $HNO_3$) and soil particles that have alkaline character (Rao et al., 2016). Mg, Al, K, Sr and Fe were negatively correlated with relative humidity (R ≤ -0.60), suggesting strong influence of drier conditions over these species.

### 3.4   Carbonaceous species and mass balance

Higher concentrations of OC and EC were observed in $Int_{2.5}$ than in $Ext_{2.5}$ with average values of 10.2 µg $m^{-3}$ for OC and 7.0

µg $m^{-3}$ for EC (Fig. 4 and Table S4). However, the difference of carbonaceous species concentrations was not considered statistically significant between the campaigns ($p$ ~0.1). The OC/EC ratios were 1.5, 1.7, and 1.8 for $Int_{2.5}$, $Ext_{2.5}$, and $Ext_{10}$, respectively. Since the ratio values were similar, as well as the absolute OC and EC concentrations were higher in intensive than extensive campaigns, this may be indicate similar sources of OC and EC are contributing all year long but with higher concentrations during $Int_{2.5}$. Ratios lower than 1 are constantly observed in roadway tunnels and are assumed to describe the

composition of fresh traffic emissions (Pio et al., 2011). Ratios ranging from 2 to 5 are commonly observed in urban background atmospheres and are assumed to indicate a significant contribution of secondary aerosol sources (Pio et al., 2011; Querol et al., 2013; Viana et al., 2007). In this way, the values for OC/EC found in the present study may be due to vehicle emissions with contribution of secondary organic aerosols.

TOM (total organic matter) was calculated by multiplying the organic carbon content by 1.6 (Timonen et al., 2013) and

represented 36, 36 and 28 % of the total PM, respectively (Fig. S2). Mass balance was determined for the aerosol considering trace elements as if they all existed as oxides (Alves et al., 2015). The unaccounted part was of 6, 15 and 26 % for $Int_{2.5}$, $Ext_{2.5}$ and $Ext_{10}$, respectively. This unaccounted part can be attributed to adsorbed water or the fact that abundant species as carbonates and Si were not determined, similarly as observed in Pio et al. (2013).

OC and EC were well correlated in $Ext_{2.5}$, with values above 0.8. This suggests that a large amount of OC is emitted by a

dominant primary source at this site (Aurela et al., 2011; Kumar and Attri, 2016). The studied site is strongly affected by vehicle emissions and, during the winter months biomass burning also contributes to these species (Pereira et al., 2017). Correlations were strong between the carbonaceous species with vehicular emitted gases such as $NO_x$ and CO (R > 0.85). OC also had good correlations with soil elements (Mg and Al) and also nss-$K^+$ (R > 0.8), suggesting association with the resuspension of road dust and also a significant biomass burning contribution.

### 355   3.4.1 Polycyclic aromatic hydrocarbons and derivatives

The PAH and derivatives concentrations are presented in Table 5. The total PAHs were higher in $Int_{2.5}$ than in $Ext_{2.5}$, 23.3 ng $m^{-3}$ and 18.4 ng $m^{-3}$, respectively (although not significantly different, with $p > 0.05$). The total PAH concentration for the $Ext_{10}$ was 24.3 ng $m^{-3}$. The lowest total PAH concentration of 2.6 ng $m^{-3}$ was observed in $Ext_{2.5}$, while the maximum of 115.3 ng $m^{-3}$ was observed in $Ext_{10}$. These levels were similar to those obtained in past studies at the same site, as 25.9 ng $m^{-3}$ in

$PM_{10}$ samples during the intensive campaign in the winter of 2008 (Vasconcellos et al., 2011a) and 27.4 ng $m^{-3}$ for $PM_{10}$ in the winter of 2003 (Vasconcellos et al., 2011b). In addition, the total PAH levels from the present study is higher than in 2013 and 2012 intensive campaigns (8.7 ng $m^{-3}$ and 8.2 ng $m^{-3}$ in $PM_{10}$) (Pereira et al., 2017). Total PAHs represented 0.23, 0.27 and 0.31 % of OC for $Int_{2.5}$, $Ext_{2.5}$, and $Ext_{10}$, respectively. In spite of accounting for a small fraction of organic carbon, it is important to observe that PAHs are among the pollutants of major concern due to their carcinogenic and mutagenic

effects.

BbF was the most abundant PAH (the BbF percentages in relation to total PAHs were 13, 12, 12 % for $Int_{2.5}$, $Ext_{2.5}$ and $Ext_{10}$, respectively) in all the campaigns. This compound has carcinogenic properties already reported in other studies (Ravindra et al., 2008). Its concentrations reached the values of 6.1, 6.4 and 13.3 ng $m^{-3}$ in $Int_{2.5}$, $Ext_{2.5}$ and $Ext_{10}$. BbF was also the most abundant PAH in 2013 intensive campaign (Pereira et al., 2017). This species was also among the most



abundant PAHs in the study performed at Jânio Quadros tunnel, with a predominance of light-duty vehicles (Brito et al., 2013). Correlations were strong between all PAHs heavier than Flt (R > 0.8); suggesting different sources from the PAHs with lower molecular weight at this site. Most of the heavier PAHs appeared to have negative correlations with temperature; the condensation of organic compounds in the aerosol is influenced by lower temperatures (Bandowe et al., 2014). Coronene, a PAH often used as a vehicular fuel marker (Ravindra et al., 2006) was correlated to vehicular related species as

Cu and Pb (R > 0.7).

BaP, the PAH most studied due to its proven carcinogenic potential, was considerably higher in $Int_{2.5}$ than in $Ext_{2.5}$. It reached the mean values of 5.5, 7.6 and 12.5 ng m$^{-3}$ in $Int_{2.5}$, $Ext_{2.5}$ and $Ext_{10}$, respectively. In the tunnels, its presence was associated with the higher contribution of LDV emissions (Brito et al., 2013). 2-NFlu and 2-NBP were among the nitro-PAHs with highest concentrations. 2-NFlu is a major component of diesel exhaust particles, such as the nitropyrenes, and is

known as a carcinogenic nitro-PAH (Draper, 1986; Fujimoto et al., 2003). 2-NFlt was moderately correlated with Flt (R = 0.4); this species is produced from reactions between Flt and $NO_2$ (Albinet et al., 2008). The ratios 2-NFlt/1-NPyr were close to 1; ratios lower than 5 indicate a predominance of primary emissions of nitro-PAHs (Ringuet et al., 2012). The compound 9,10-AQ was the most abundant oxy-PAH found in this study. It can be either primarily emitted or secondarily formed. A recent study showed that it can be formed from the heterogeneous reaction between $NO_2$ and Ant adsorbed on NaCl particles

(sea salt) (Chen and Zhu, 2014). A moderate correlation was found between 9,10-AQ and Ant (R = 0.54).

### 3.4.2 PAH diagnostic ratios

The PAH diagnostic ratios (Table 5) were obtained for all the campaigns, since they can point to some emission sources, such as oil products, fossil fuels, coal or biomass combustion. However, these ratios should be used with caution, due to the peculiarity of fuel compositions in Brazilian's car fleet. The values of PAH ratios can also be affected by changes of phase,

transport and degradation (Tobiszewski and Namieśnik, 2012).The ratio BaP/(BaP+BeP) is related to the aerosol photolysis. Most of the local PAH emissions contain equal concentrations of BeP and BaP. However, BaP is more likely to undergo photolysis or oxidation (Oliveira et al., 2011). The average BaP/(BaP+BeP) was close to 0.4 for the three campaigns. This ratio was slightly lower than the ratio obtained in the 2013 intensive campaign, although still very close to 0.5 (Pereira et al., 2017); it is suggested that the PAHs found in the site are mostly emitted locally.

The Flt/(Flt+Pyr) and InP/(InP+BPe) ratios were reported to be the most conservative by Tobiszewski and Namieśnik (2012). The Flt/(Flt+Pyr) ratio for all the campaigns were close to 0.5, falling within the range for fossil fuel combustion (0.4–0.5) (de la Torre-Roche et al., 2009). The ratio InP/(InP+BPe) represented values close to 0.5, similar to the ratio obtained for JQ tunnel (0.55) impacted by LDV (the ratio found for MM tunnel was 0.36). The average BaA/Chr ratio ranged between 0.5–0.6 in the 2014 campaigns, also approaching that of JQ tunnel (0.48) (Brito et al., 2013), whilst a value

of 0.79 was obtained for MM tunnel. The BaA/(BaA+Chr) ratio was reported to be sensitive to photodegradation (Tobiszewski and Namieśnik, 2012). However, it is possible to consider that this degradation was not significant due to proximity to the emission sources (the expressway). All ratios suggested a greater contribution of LDV to PAHs at the sampling site.

The ΣLMW/ΣHMW ratios (LMW - Low Molecular Weight PAHs with three and four aromatic rings and HMW - High

Molecular Weight PAHs with more than four rings) were considerably low in all campaigns (predominance of HMW PAHs). It is known that LMW PAHs have higher concentrations in the gas phase while HMW PAHs are preferentially present in PM (Agudelo-Castañeda and Teixeira, 2014; Duan et al., 2007). The HMW PAHs contribution was higher in winter, just as the ratios were lower, corroborating the results of some previous studies (Chen et al., 2016; Teixeira et al., 2013). In turn, HMW PAHs are more likely to be retained in particles due to its lower vapor pressure than LMW PAHs.

LMW PAHs are also mostly associated with diesel engines, while HMW PAHs are predominantly emitted by gasoline exhaust (Chen et al., 2013; Cui et al., 2016; Miguel et al., 1998). The ratio between BPe and BaP for all campaigns (all close





to 1) was very similar to that found in a study with Brazilian light-duty diesel vehicles exhaust (1.13) (de Abrantes et al., 2004) and also found in a campaign performed in São Paulo (1.11) (de Martinis et al., 2002). This may be a characteristic fingerprint for local vehicular emissions.

### 3.4.3 PAHs risk assessment

BaPE is a parameter introduced to quantify the aerosol carcinogenicity related to all carcinogenic PAHs instead of BaP solely. BaPE values above 1.0 ng m$^{-3}$ represent an increased cancer risk. The carcinogenic nitro-PAHs (1-NPyr, 4-NPyr, 6-NChr) were below the detection limit in most part of the extensive campaign samples, so they were not considered in the risk assessment. BaPE is calculated according to Eq. (5), given by Yassaa et al. (2001) and Vasconcellos et al. (2011a) :

$$BaPE = ([BaA] \times 0.06) + ([BbF] \times 0.07) + ([BkF] \times 0.07) + ([BaP] \times 1) + ([DBA] \times 0.6) + ([InP] \times 0.08) \quad (5)$$

The BaPE values for the $Int_{2.5}$ ranged between 0.6 and 8.0 ng m$^{-3}$ and for $Ext_{2.5}$, between 0.3 and 10.5 ng m$^{-3}$, while the average of BaPE for the $Int_{2.5}$ was considerably higher than in $Ext_{2.5}$ (3.4 and 2.4 ng m$^{-3}$, respectively). In the $Ext_{10}$ this index ranged between 0.5 and 18.3 ng m$^{-3}$. The maximum value was even higher than the value of 12.1 ng m$^{-3}$ in $PM_{10}$, obtained in São Paulo in an intensive campaign conducted in 2008 (Vasconcellos et al., 2011a). More than 70 % of the samples in the

$Ext_{10}$ had BaPE indexes higher than 1 ng m$^{-3}$. The year 2014 was a relatively dry year, with an annual rainfall 13 % below the average (IAG, 2014). The average values for BaPE in $PM_{10}$ at the site were 1.9 and 3.7 ng m$^{-3}$ in the intensive campaigns of 2007 and 2008, respectively. On the other hand, at forested areas in São Paulo state the value can be as low as 0.1 ng m$^{-3}$ (Vasconcellos et al., 2010).

The lifetime lung cancer risks (LCR) were assessed from the carcinogenic potential (BaP-TEQ) and mutagenic potential

(BaP-MEQ ), through equations (6) and (7) (Jung et al., 2010):

$$(BaP - TEQ) = ([BaA] \times 0.1) + ([Chr] \times 0.01) + ([BbF] \times 0.1) + ([BkF] \times 0.1) + ([BaP] \times 1) + ([InP] \times 0.1) +$$
$$([DBA] \times 5) + ([BPe] \times 0.01) \quad (6)$$

$$(BaP - MEQ) = ([BaA] \times 0.082) + ([Chr] \times 0.017) + ([BbF] \times 0.25) + ([BkF] \times 0.11) + ([BaP] \times 1) +$$
$$([InP] \times 0.31) + ([DBA] \times 0.29) + ([BPe] \times 0.19) \quad (7)$$

LCR from exposure to atmospheric PAH was estimated by multiplying BaP-TEQ and BaP-MEQ by the unit risk ($87 \times 10^{-6}$ (ng m$^{-3}$)$^{-1}$) for exposure to BaP established by WHO (de Oliveira Alves et al., 2015; WHO, 2000) (Fig. 5) and was possible to observe an increase during the intensive campaign.

### 3.5 Biomass burning tracers

The highest concentrations of biomass burning tracers (levoglucosan, means 509 ng m$^{-3}$; mannosan 45 ng m$^{-3}$; galactosan 33

ng m$^{-3}$) were observed in the $Int_{2.5}$ ($p \sim 0.05$), during the biomass burning period (Fig. 6 - Table S5). In the intensive campaign period, 1364 fire spots were registered in São Paulo state, with an average of 72 fires per day (INPE, 2014). In the same way, 65 % of the sampling days the backward air masses passed through regions with biomass burning. The average concentration of levoglucosan obtained in the $Int_{2.5}$ (509 ng m$^{-3}$) were higher than those of the intensive $PM_{10}$ sampling campaigns in 2013 and 2012 (474 ng m$^{-3}$ and 331 ng m$^{-3}$) (Caumo et al., 2016; Pereira et al., 2017), as well as more than

twice than the values obtained in the 2008 intensive campaign (Vasconcellos et al., 2010).

The Lev/Man ratios are characteristic of each type of biomass. The ratios were similar to that obtained in a chamber study with sugarcane burning in Florida (Lev/Man = 10) (Hall et al., 2012), and also to that reported for the 2013 intensive campaign (Lev/Man = 12) (Pereira et al., 2017). Nss-K$^+$/Lev ratios were 1.6, 1.4 and 1.3 for $Int_{2.5}$, $Ext_{2.5}$ and $Ext_{10}$, respectively. These ratios are similar to those obtained in the previous $PM_{10}$ intensive campaign (1.4) in 2013, which was

attributed to a combination of smouldering (flameless combustion) and flaming processes during the combustion of biomass





(Kundu et al., 2010; Pereira et al., 2017). The flaming combustion is predominant for sugarcane leaves (Hall et al., 2012; Urban et al., 2016).

Correlations between potassium and monosaccharides in $Ext_{2.5}$ were high (R > 0.8), indicating that, most of the year, potassium in $PM_{2.5}$ can be linked to biomass burning. Coarse fraction potassium, more related to soil sources, did not present
strong correlations with levoglucosan. Local burning also can affect the site since some restaurants consume wood for cooking (pizzerias and steakhouses) (Kumar et al., 2016). There is a stronger correlation between chloride and other biomass burning tracers in $Ext_{2.5}$ than in $Ext_{2.5-10}$. Chloride is also a major emission from biomass burning, in the form of KCl (Allen et al., 2004) and is also emitted as HCl in garbage burning (Calvo et al., 2013). Carbonaceous species presented high correlations (R > 0.75) with levoglucosan in $Ext_{2.5}$. This suggests that some of these species may be also linked to
biomass burning emissions.

During the intensive campaign, the backward trajectories pointed to air masses passing by regions affected by biomass burning on 65 % of the sampling days. The highest concentrations of biomass burning tracers were found on 1[st] of July, when the levoglucosan level reached 1263 ng m$^{-3}$. On that day, about 100 fire spots (INPE, 2014) were observed in the state of São Paulo and the back trajectories revealed air masses crossing the West and Northwest of the state (Fig. 7a), where the
fire spots were observed. In this same sampling day, local fire spots were observed, possibly landfill burning.

On 12[th] of July, the air masses travelled through the Atlantic Ocean before reaching the site. In the same period, the $PM_{2.5}$ and biomass burning tracers concentrations dropped. Figure 7b shows the trajectories for 13[th] of July. Some of the lowest concentrations of levoglucosan (80 and 74 ng m$^{-3}$) and $PM_{2.5}$ (28 and 26 µg m$^{-3}$) were observed on 12[th] and 13[th] of July, respectively.

**3.6 Distribution of species in fine and coarse particles during extensive campaigns**

Figure 8 shows the mass percentage of tracers in fine ($PM_{2.5}$) and coarse particles ($PM_{2.5-10}$) in the extensive campaign; their values are presented in SI (Table S6). The biomass burning tracers, levoglucosan and mannosan were present mostly in $PM_{2.5}$ mass fractions (over 75 %). In this study, 73 % of nss-$K^+$ mass was in $PM_{2.5}$. This species may also be attributed to biomass burning, although coarse potassium may be from soil dust resuspension (Souza et al., 2014a; Vasconcellos et al.,
2011a).

Species related to vehicular emissions as coronene and Cu (Brito et al., 2013; Ravindra et al., 2006) were also predominantly found in $PM_{2.5}$ (73 and 61 %, respectively). More than 50 % of Fe and Ca, crustal elements, was found in $PM_{2.5-10}$. A previous source apportionment study in Southern European cities (AIRUSE-LIFE+ project) also pointed out soil dust as a significant source, accounting for 2–7 % of $PM_{2.5}$ at suburban and urban background sites and 15 % at a traffic impacted
station. In the case of $PM_{10}$, these percentages increased to 7–12 % and 19 %, respectively (Amato et al., 2016).

In a previous winter campaign in 2008, in São Paulo (Souza et al., 2014a) levoglucosan and mannosan were also mostly present in $PM_{2.5}$. Urban et al. (2014) found for an agro-industrial region in São Paulo state that between 58 and 83 % of levoglucosan was present in particles smaller than 1.5 µm. It is similar to the values observed in other studies done in the state of São Paulo and in the Amazon region (Decesari et al., 2006; Schkolnik et al., 2005; Urban et al., 2012).

Sulfate and ammonium were predominant in $PM_{2.5}$ (over 65% and 80%). Sulfate was also predominant in $PM_{2.5}$ in a previous study done in São Paulo between 1997 and 1998; it was attributed to the gas-to-particle conversion of vehicular $SO_2$ (Castanho and Artaxo, 2001). Both ions may be present as $(NH_4)_2SO_4$ in the fine mode. Nitrate is well distributed in both phases, likely resulting from reactions of $HNO_3$ with soil species (Tang et al., 2016). In turn, fine mode nitrate is often present in the form of ammonium nitrate, which is a thermally unstable species (Maenhaut et al., 2008).

Other ions, such as sodium and chloride were halved in each mode. These species are related to sea salt aerosols and are more often present in the coarse mode, as observed in the study done at urban sites in Rio de Janeiro (Godoy et al., 2009). Chloride in $PM_{2.5}$ can also be originated by biomass burning emissions (Allen et al., 2004). OC and EC, which are mainly





related to vehicular emissions in São Paulo (Castanho and Artaxo, 2001), were mostly in the fine particles. OC and EC were also associated with biomass burning in a recent study (Pereira et al., 2017).

BaP was found mainly in the $PM_{2.5}$ (over 80 %), which are able to be deposited in the tracheobronchial region of the human respiratory tract, representing an increased health risk (Sarigiannis et al., 2015). In turn, As, Cd and Pb, identified as elements that can cause carcinogenic health effects (Behera et al., 2015), were also found predominantly in the $PM_{2.5}$ (over 75 %). In this way, they may be indicative of higher carcinogenicity of fine over coarse particles.

### 3.7.  Source apportionment by PMF and polar plots

Source apportionment was performed with PMF including all data. Then, the factor contributions were separated for each campaign ($n = 78$). Eleven *strong* species were considered ($SO_4^{2-}$, nss-$K^+$, Mg, Cr, Mn, Fe, Ni, Cd, Pb, OC and EC), six were considered *weak* (Lev, Man, $NO_3^-$, $NH_4^+$, Ca and Cu) and the PM concentrations were set as a total variable. An extra modelling uncertainty of 25 % was added to the model.

Considering the limited number of samples, a restricted number of species had to be chosen. Elements already studied and
attributed to sources in São Paulo were preferred.  In some of the base model runs it was possible to observe a sea salt profile with $Na^+$ and $Cl^-$, but after they were removed, other profiles were clearly improved. PAHs were firstly included in the model but it created a factor associated with temperature conditions, increasing in the dry season since the lower dispersion conditions in the period favor the accumulation of HMW-PAHs in suspended particles (Agudelo-Castañeda and Teixeira, 2014; Ravindra et al., 2006) Lev and man were set as *weak* due to their organic character, and $NO_3^-$ and $NH_4^+$ due to their
thermal instability. Ca and Cu also had to be set as *weak* in order to have a convergent base model run.

Solutions with three to eight factors were tested. The ratio of robust to theoretical parameters ($Q_R/Q_T$) reduced between simulations when increasing the number of factors. A solution with five factors was found to have more meaningful results; $Q_R$ and $Q_T$ values were 367 (Table S7). The source profiles obtained in the PMF analysis and the contribution of each factor to $PM_{10}$ concentrations are found in Fig. 9. Constraints were applied, Cu was pulled up maximally in the vehicular factor and
Lev and Man were pulled up maximally in the biomass burning factor in order to have a better separation between both factors. The PMF result charts are presented in Fig. S3.

Factor 1 presented higher loadings for Mg, Ca and Fe, elements associated with soil resuspension in previous studies (da Rocha et al., 2012). The factor was also mixed with vehicular related species, such as Cu and OC, which can be attributed to the resuspension of road dust by traffic. Accounting for 24.3, 12.5 and 25.7 % of $Int_{2.5}$, $Ext_{2.5}$ and $Ext_{10}$, respectively, it was
the most important source for the $PM_{10}$ campaign. In some runs, it was possible to observe Li and Tl in this factor, but these species were not considered in the final model. This soil contribution was similar to that obtained for $PM_{10}$ in a year round inventory in the city (CETESB, 2015). High loadings for ions, such as nss-$K^+$ and $NO_3^-$, were also present in the factor. Gaseous $HNO_3$ can interact with soil particles and form coarse nitrates (Tang et al., 2016). The factor contribution appeared to increase with wind speed from NW and decrease with SE winds (Fig. 10). Soil dust and vegetation sources also tended to
reduce with SE winds, as observed previously by Sánchez-Ccoyllo and Andrade (2002).

Factor 2 shows high loads for Ni, Pb and Cr, which are often attributed to industrial emissions (Bourotte et al., 2011; Castanho and Artaxo, 2001). This factor had some of the lowest contributions, 10.5, 9.7 and 9.5 % for $Int_{2.5}$, $Ext_{2.5}$ and $Ext_{10}$ and appeared to increase with SE winds, passing through nearby industrial regions (southeast of the city). The growth of industries have been limited in the last years and vehicle fleet is expected to be a main source of atmospheric pollutants in
the area (Kumar et al., 2016).

Factor 3 showed high loadings for vehicular related tracers, such as Cu, Fe, OC and EC (with a higher load on EC and Cu). Cu and Fe were found in the LDV impacted tunnel study in São Paulo and Cu is emitted from brake pads, in stop-and-go driving in the expressways (Andrade et al., 2012b; Brito et al., 2013) and also are present in ethanol after the processing of copper tanks. Loading still relatively high in this were observed for levoglucosan and mannosan, which precluded the total





separation of biomass burning tracers. On days with NW winds, both source contributions tended to increase as observed in the polar plots. This factor represented 30.9, 39.1 and 39.2 % contribution for Int$_{2.5}$, Ext$_{2.5}$ and Ext$_{10}$, and had a constant contribution comparing dry and wet period in the Ext$_{10}$ campaign. Vehicular source seemed to increase with winds coming from the North and Northwest, passing by the expressway, but decreased with SE winds, as observed previously (Sánchez-Ccoyllo and Andrade, 2002).

Factor 4 was associated with biomass burning due to the loadings for levoglucosan, mannosan and non-sea-salt potassium, OC and EC. It is also noteworthy the loading of Cd in this factor; wood burning was pointed out as a possible source of this metal in a previous study in Belgium (Maenhaut et al., 2016), more studies necessary in order to explain the biomass burning contribution to this species in São Paulo. This factor represented 18.3, 11.6 and 7.6 % for Int$_{2.5}$, Ext$_{2.5}$ and Ext$_{10}$, respectively. The contributions of this factor were higher in the intensive campaign (sugarcane burning period), but were also present in

the other periods, suggesting other biomass burning sources in the city, such as waste burning and wood stoves (Kumar et al., 2016). Several fire spots were registered in São Paulo state in the intensive campaign, some of them in the neighboring towns (INPE, 2014). The polar plot showed that this factor tended to increase with NW winds, passing through the inland of São Paulo state, and decrease with SE (more humid) winds from the sea.

Factor 5 was attributed to the secondary inorganic aerosol formation processes (loadings for NO$_3^-$, SO$_4^{2-}$ and NH$_4^+$) and also

OC (secondary organic carbon). The contributions were 15.9, 27.1 and 17.9 % for Int$_{2.5}$, Ext$_{2.5}$ and Ext$_{10}$, respectively. The contributions of this profile did not follow any seasonal trend (Fig. S3a). In 2014, 78 % of NO$_x$ and 43 % of SO$_x$ emissions in Greater São Paulo were attributed to the vehicle fleet (CETESB, 2015). Taking into account that in São Paulo SO$_x$ and NO$_x$ concentrations are similar all year round, this could explain the lack of seasonality of this factor. The polar plot showed a centralized profile, increasing with lower wind speed, which suggests a local secondary process.

Other polar plots were obtained for individual species and are presented in Fig. 11. It is possible to see that Na$^+$ tended to increase with stronger winds coming from the sea, while Cl$^-$ had a different pattern. Chloride in the marine aerosol can be depleted after atmospheric reactions with acids (Calvo et al., 2013; White, 2008). It is noteworthy that MSA was associated with NW winds, This species is often associated with the decomposition of DMS, emitted by the sea (Bardouki et al., 2003). More studies are needed in order to identify MSA sources on this site. Similarly, as for biomass burning factor, levoglucosan

tended to increase with NW winds. However, it is also possible to observe local sources for this species due to its high levels, even with lower speed wind. Secondarily formed species such as NO$_3^-$ and SO$_4^{2-}$ had a much-centered profile and tended to increase with lower wind speed. EC, Chr and Cor seemed to be emitted by local sources, likely vehicular emissions (Alves et al., 2016; Ravindra et al., 2008). On the other hand, Flt (a light molecular mass PAH), seemed to be influenced by different air masses, suggesting different sources.

**4    Summary and conclusions**

Particulate matter (PM$_{2.5}$ and PM$_{10}$) was collected throughout the year 2014 to determine different chemical constituents, including carbonaceous species, WSI, monosaccharides, PAHs, and their derivatives. The risks of PAHs for human health were assessed with levels exceeding the suggested guidelines. Higher concentrations of biomass burning species were found in the fine particles during the campaigns. Good correlations were found between the monosaccharides and OC and EC,

highlighting their contributions to carbonaceous species. Non-sea-salt potassium was also well correlated with the biomass burning species, corroborating the input from this source.

PMF analysis was performed and source profiles were obtained for Int$_{2.5}$, Ext$_{2.5}$ and Ext$_{10}$. Five factors were identified: road dust, industrial, vehicular, biomass burning and secondary processes. Almost 20 % of biomass burning contribution was observed for the PM$_{2.5}$ intensive sampling campaign. The source apportionment led to the identification of traffic-related

sources, as expected for the site, since the samples were collected during weekdays. The considerable biomass burning contribution suggests not only the importance of long-range transport of emissions from sugarcane burning, but also the





input from local biomass burning sources, such as waste burning and wood stoves in restaurants. More studies are needed on the impact of local sources of biomass burning, in order to identify the different inputs.

**Data availability:** All data are available by the authors Guilherme Martins Pereira (martinspereira2@hotmail.com) and
Pérola de Castro Vasconcellos (perola@iq.usp.br). Meteorological data can be found at CETESB, 2014, bulletin (http://www.estacao.iag.usp.br/Boletins/2014.pdf). Gaseous species concentration can be found at CETESB (http://sistemasinter.cetesb.sp.gov.br/Ar/php/ar_dados_horarios.php).

**Author contributions:** Mr. Guilherme M. Pereira performed the PAHs, WSI and monosaccharides determination and the PMF analysis and Prof. Dr. Pérola Vasconcellos is his advisor and laboratory supervisor in Brazil. Dr. Kimmo Teinillä was
responsible for WSI and monosaccharides determination with Mr. Guilherme Pereira, and Dr. Risto Hillamo is the laboratory supervisor. Dr. Danilo Custódio was responsible for carbonaceous species determination and polar plots, Dr. Célia Alves is the department supervisor in Portugal. Dr. Aldenor Gomes Santos developed the method for PAHs determination; Prof. Gisele Olímpio da Rocha and Prof. Jailson Bittencourt de Andrade are the supervisors at UFBA. Prof. Prashant Kumar and Prof. Maria de Fátima Andrade contributed with source apportionment. Ms. Huang Xian and Prof.
Rajasekhar Balasubramanian were responsible for elements determination.

**Competing interests:** The authors declare that they have no conflict of interest.

**Acknowledgements:** This work was supported by grants from FAPESP, São Paulo Research Foundation and CNPq (Project 152601/2013-9), National Council for Scientific and Technological Development, for the postgraduate scholarship and Santander Bank, for international scholarship in Helsinki, Finland. The authors also thank the INCT- Energy and
Environment. Prof. Prashant Kumar also acknowledges the collaborative funding received to Universities of Surrey and São Paulo through the UGPN funded projects BIOBURN (Towards the Treatment of Aerosol Emissions from Biomass Burning in Chemical Transport Models through a case study in the Metropolitan Area of São Paulo) and NEST-SEAS (Next-Generation Environmental Sensing for Local to Global Scale Health Impact Assessment) that allowed Guilherme Martins Pereira to work at the University of Surrey, United Kingdom. Prof. P. C. Vasconcellos, G. O. da Rocha and J. B. de Andrade
thank CNPq for their fellowships. A. G. Gomes, G. O. da Rocha and J. B. de Andrade also thank CAPES, CNPq, FAPESB, FINEP and PETROBRAS for research funding at UFBA. Finally, G. O. da Rocha is thankful for a partial fellowship funding from Fundação Lehmann. Mr. Guilherme Pereira also thanks Dr. Ioar Rivas and students Bruna Segalin and Fatima Khanun for the help with the PMF analysis.

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




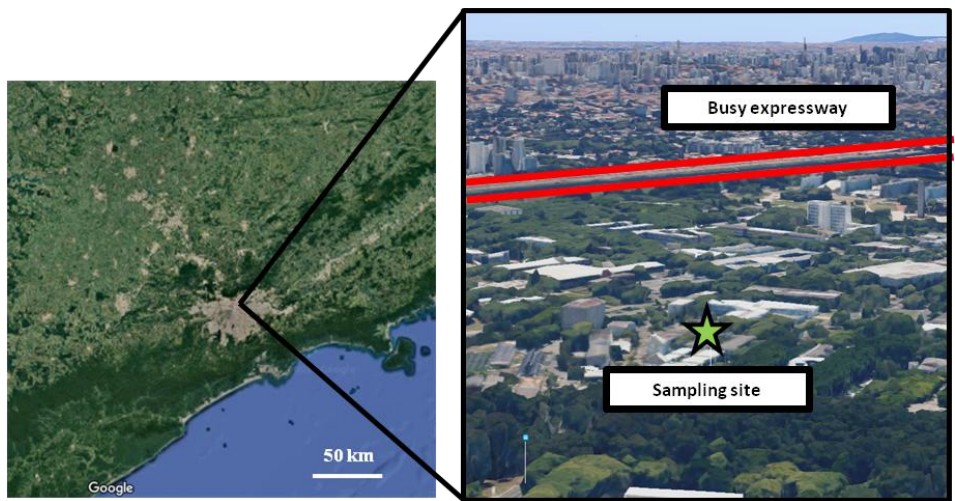

**Figure 1. Location of the sampling site. Maps are a courtesy of Google maps.**



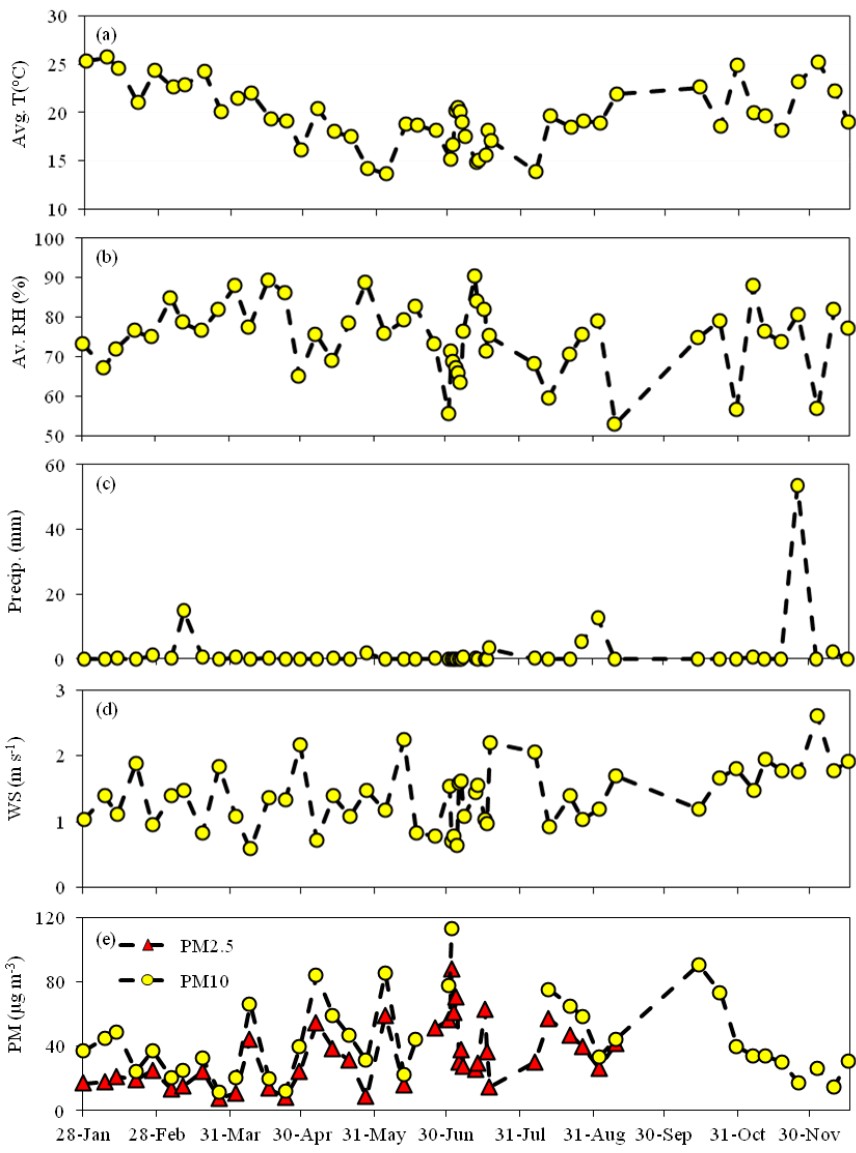

**Figure 2. (a) Daily average temperatures, (b) relative humidity (RH), (c) precipitation, (d) wind speed and (e) PM$_{10}$ and PM$_{2.5}$ concentrations for the three campaigns.**





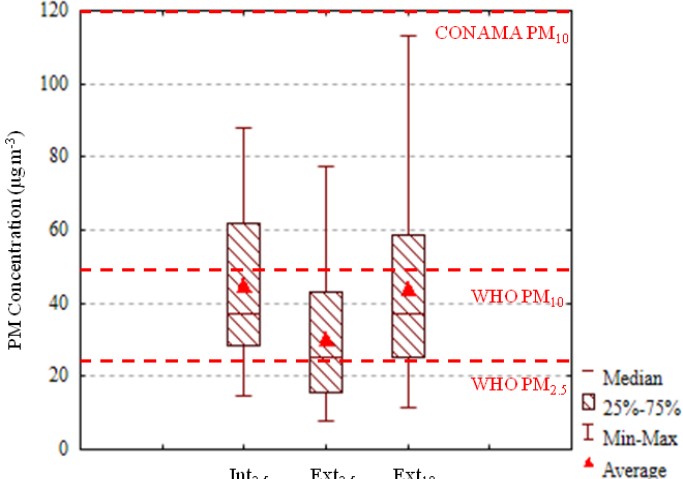

**Figure 3. Box plot for particulate matter concentrations in the intensive and extensive campaigns.**

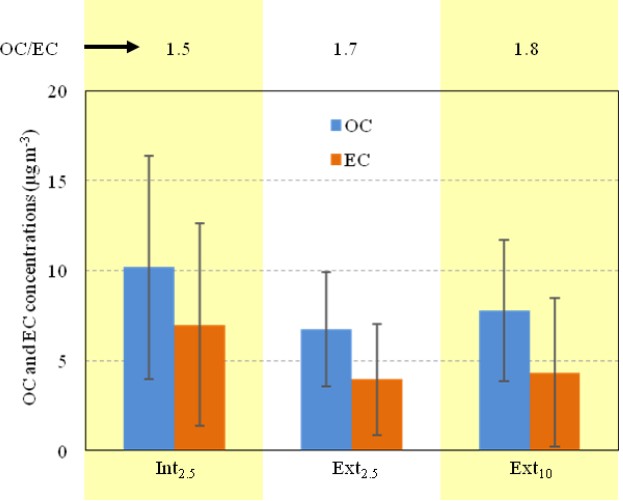

**Figure 4. Carbonaceous species concentrations for all campaigns.**




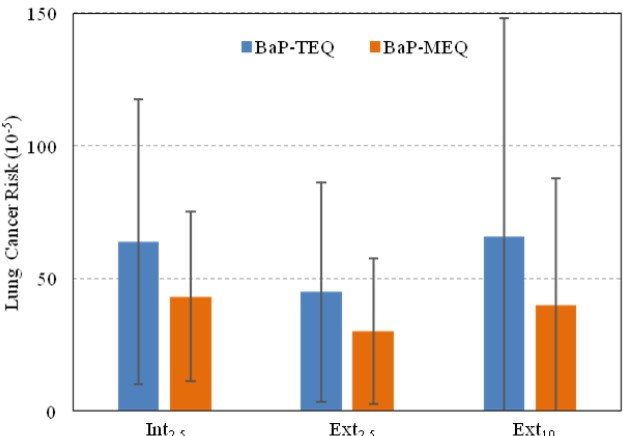

**Figure 5**. **Lung cancer risk from the BaP-TEQ and BaP-MEQ for all the three campaigns.**

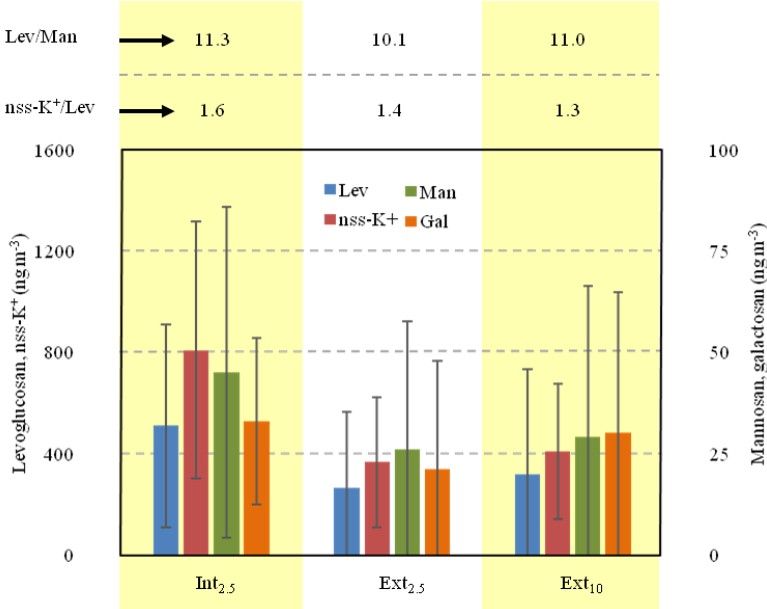

**Figure 6. Concentrations of biomass burning tracers for all campaigns.**




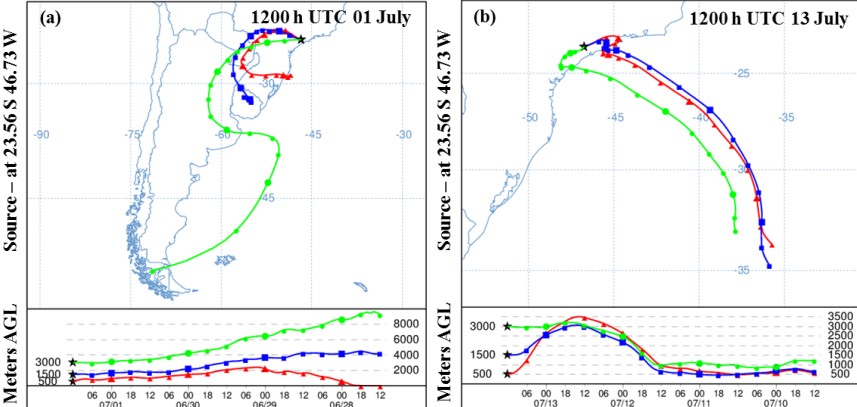

**Figure 7. Backward air mass trajectories for the days (a) 01 July and (b) 13 July.**


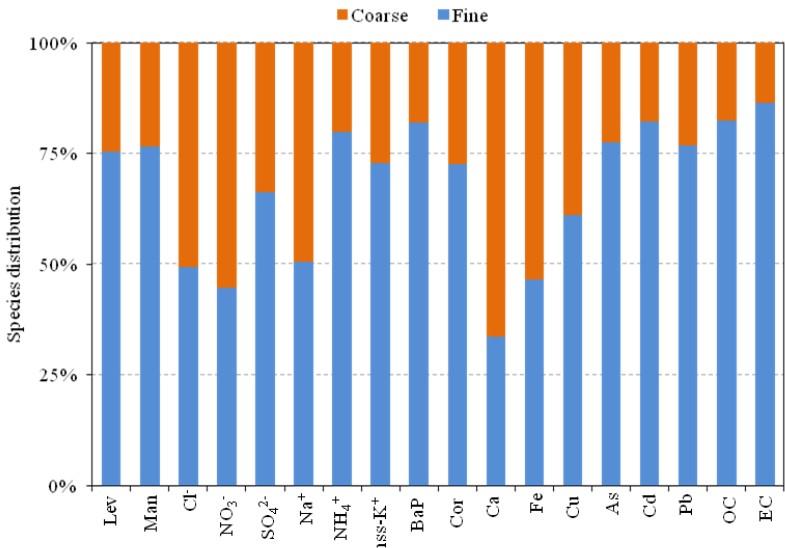

**Figure 8**. **Mass percentage distribution of species in the fine and coarse particles.**





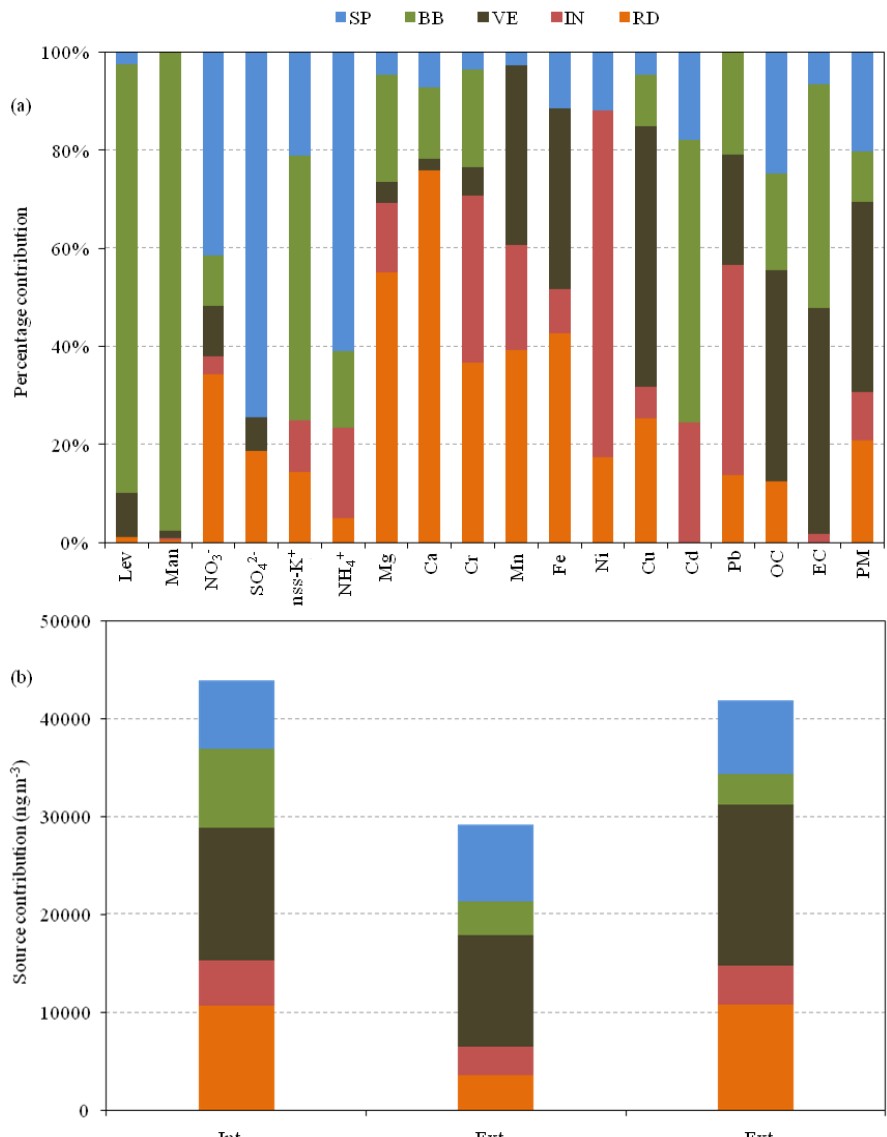

**Figure 9. (a) Profile of species for each source (SP - Secondary processes, BB - Biomass burning, VE - Vehicular, IN - Industrial and RD - Road dust) (b) Contribution of sources for each campaign.**



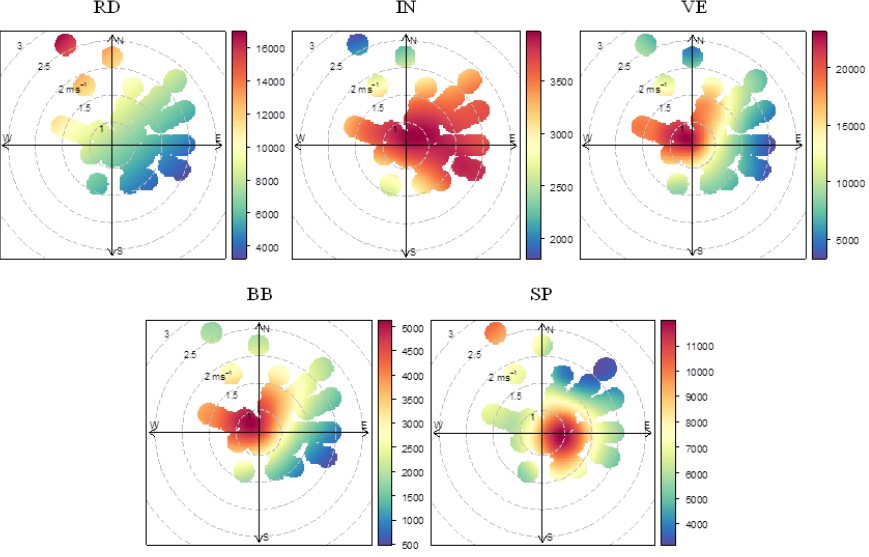

Figure 10. Polar plots of sources contributions in São Paulo (ng m$^{-3}$ and m s$^{-1}$).





**Figure 11. Polar plots for different species (ng m$^{-3}$ and m s$^{-1}$).**




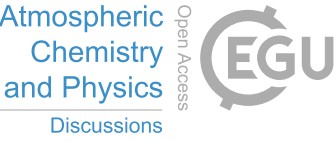

**Table 1. Results of previous source apportionment studies in São Paulo city.**

| Site | Year | Instruments | Species | Range | Identified Sources | Source |
|---|---|---|---|---|---|---|
| University of São Paulo - Atmospheric Sciences Department (campus) | 1989 (winter) | PIXE | Elements | PM$_{2.5}$ | Factor analysis - five sources: industrial emissions (13 %), emissions from residual oil and diesel (41 %), resuspended soil dust (28 %) and emissions of Cu and Mg (18 %). | Andrade et al. (1994) |
| | | | | PM$_{2.5-15}$ | Four sources: soil dust (59%), industrial emissions (19 %), oil burning (8 %) and sea salt aerosol (14 %). | |
| University - Medicine School (downtown) | 1994 (winter) | PIXE, reflectance | Elements, black carbon | PM$_2$ | Factor analysis - five sources: vehicles, garbage incineration, vegetation, suspended soil dust and burning of fuel oil. | Sánchez-Ccoyllo and Andrade (2002) |
| University - Medicine School building and Atmospheric Sciences Department | 1997 (winter) and 1998 (summer) | TEOM; PIXE; ACPM; Aethalometer | OC, EC, elements and gaseous species | PM$_{2.5}$ | Factor analysis - five sources: motor vehicle (28 and 24 %, for winter and summer), resuspended soil dust (25 and 30 %), oil combustion source (18 and 21 %), sulfates (23% and 17%) and industrial emissions (5% and 6%). | Castanho and Artaxo (2001) |
| | | | | PM$_{2.5-10}$ | Resuspended soil dust represented a large fraction (75–78 %). | |
| University of São Paulo - Atmospheric Sciences Department | 2002 (winter) | GC-MS | PAHs | PM$_{2.5}$ | Factor analysis - four factors: diesel emissions, stationary combustion source, vehicular emissions, natural gas combustion and biomass burning. | Bourotte et al. (2005) |
| | | | | PM$_{2.5–10}$ | Two factors: vehicular emissions and mixture of combustion sources (natural gas combustion, incineration emissions and oil combustion). | |
| University of São Paulo - Atmospheric Sciences Department | 2003 (winter) | IC; ICPMS | WSI and Elements | PM$_{10}$ | Principal component analysis – Two factors (48.5 % of variance): local and remote sources. | Vasconcellos et al. (2007 |
| University of São Paulo - Atmospheric Sciences Department | 2003-2004 (year round) | IC; CCD ICP | WSI and Elements | PM$_{10}$ | Principal component analysis - three principal components: biomass burning and/or automobile fuel burning (40.3 %), gas-to-particle | da Rocha et al., (2012) |





| | | | | | | |
|---|---|---|---|---|---|---|
| University - Medicine School building | 2007-2008 (year round) | X-ray spectrometry, reflectance | Elements and black carbon | PM$_{2.5}$ | conversion (12.7 %) and sea spray contribution (11.7 %). APCA - four factors: crustal emission (soil and construction) (13 %); oil-burning boilers, industrial emissions and secondary aerosol formation (13 %); light-duty vehicle emissions (12 %) and heavy-duty diesel fleet (28 %). | Andrade et al. (2012) |













**Table 2. Details of the analyzed species, analytical methods and detection limits.**

| Analytical method | Detection limits (ng m$^{-3}$) | Determined species |
|---|---|---|
| **Thermal-optical analysis** | 14 (EC) / 262 (OC) | **Carbonaceous species:** OC and EC. |
| **GC-MS** | 0.01–0.06 | **PAHs:** naphthalene (Nap), acenaphthene (Ace), acenaphthylene (Acy), Fluorene (Flu), phenanthrene (Phe), anthracene (Ant), fluoranthene (Flt), pyrene (Pyr), benzo[a]anthracene (BaA), chrysene (Chr), benzo[b]fluoranthene (BbF), benzo[k]fluoranthene (BkF), benzo[e]pyrene (BeP), benzo[a]pyrene (BaP), perylene (Per), indeno[1,2,3-cd]pyrene (InP), dibenzo[ah]anthracene (DBA), benzo[ghi]perylene (BPe) and coronene (Cor). |
| | 0.01–0.50 | **Nitro-PAHs:** (1-nitronaphthalene (1-NNap), 2-nitronaphthalene (2-NNap), 1-methyl-4-nitronaphthalene (1-Methyl-4-NNap), 1-methyl-5-nitronaphthalene (1-Methyl-5-NNap), 1-methyl-6-nitronaphthalene (1-Methyl-6-NNap), 2-methyl-4-nitronaphthalene (2- methyl-4-NNap), 2-nitrobiphenyl (2-NBP), 3-nitrobiphenyl (3-NBP), 4-nitrobiphenyl (4-NBP), 5-nitroacenaphthene (5-NAce), 2-nitrofluorene (2-Nflu), 2-nitrophenanthrene (2-NPhe), 3-nitrophenanthrene (3-NPhe), 9-nitrophenanthrene (9-NPhe), 2-nitroanthracene (2-NAnt), 9-nitroanthracene (9-NAnt), 2-nitrofluoranthene (2-NFlt), 3-nitrofluoranthene (3-NFlt), 1-nitropyrene (1-NPyr), 2-nitropyrene (2-NPyr), 4-nitropyrene (4-NPyr), 6- nitrochrysene (6-NChr), 7-nitrobenz[a]anthracene (7-NBaA), 3-nitrobenzanthrone (3- NBA), 6-nitrobenzo[a]pyrene (6-NBaPyr), 1-nitrobenzo[e]pyrene (1-NBePyr), and 3-nitrobenzo[e]pyrene (3-NBePyr). |
| | 0.3–10.3 | **Oxy-PAHs:** 1,4-benzoquinone (1,4-BQ), 9,10-phenanthraquinone (9,10-PQ), 9,10-anthraquinone (9,10-AQ), 1,2-naphthoquinone (1,2-NQ) and 1,4-naphthoquinone (1,4-NQ). |
| **IC** | 6.4–6.4 | **WSI:** Cl$^-$, NO$_3^-$, SO$_4^{2-}$, C$_2$O$_4^{2-}$, methylsulfonate (MSA$^-$), Na$^+$, K$^+$, NH$_4^+$. |
| **HPAEC-MS** | 1.3–2.5 | **Monosaccharides:** levoglucosan, mannosan and galactosan. |
| **ICP-MS** | 0.0002–2.3 | **Elements:** Li, Mg, Al, K, Ca, Cr, Mn, Fe, Co, Ni, Cu, Zn, As, Se, Rb, Sr, Cd, Sn, Cs, Tl, Pb, Bi. |

**Table 3. Concentrations of WSI in all campaigns.**

| (ng m$^{-3}$) | Int$_{2.5}$ Average (Min–Max) | Ext$_{2.5}$ Average (Min–Max) | Ext$_{10}$ Average (Min–Max) |
|---|---|---|---|
| **Cl$^-$** | 964 (107–4549) | 330 (16–1427) | 641 (76–5904) |
| **NO$_3^-$** | 2678 (667–6873) | 1430 (183–3419) | 2872(437–8880) |
| **SO$_4^{2-}$** | 3266 (1252–5959) | 3197 (922–6300) | 3680 (569–9361) |
| **MSA$^-$** | 84 (15–214) | 63 (13–226) | 107 (28–444)* |
| **C$_2$O$_4^{2-}$** | 478 (176–753) | 282 (57–726) | 367 (50–1180) |
| **Na$^+$** | 350 (46–869) | 238 (64–512) | 571 (76–1908) |
| **NH$_4^+$** | 1712 (613–4075) | 1370 (281–2845) | 1336 (57–4436) |
| **nss-K$^+$** | 809 (237–2007) | 366 (49–1137) | 413 (63–1181) |
| **SNA** | 7655 | 5997 | 7888 |
| **Total** | 10334 | 7276 | 9986 |
| **SO$_4^{2-}$/NO$_3^-$** | 1.2 | 2.2 | 1.3 |
| **Cl$^-$/Na$^+$** | 2.7 | 1.4 | 1.1 |
| **∑cations/∑anions** | 0.88 | 0.58 | 0.64 |
| **SNA/Total (%)** | 74 | 82 | 79 |

*Data was not determined after 21 August.*





**Table 4. Average, minimum and maximum concentrations of tracer elements for all campaigns.**

| (ng m$^{-3}$) | Int$_{2.5}$ Average (Min–Max) | Ext$_{2.5}$ Average (Min–Max) | Ext$_{10}$ Average (Min–Max) |
|---|---|---|---|
| **Li** | 0.48 (ND –1.12) | 0.27 (ND –0.70) | 0.40 (ND–1.25) |
| **Mg** | 210 (5–469) | 93 (5–356) | 154 (ND–377) |
| **Al** | 1851 (ND –2782) | 691 (ND –2712) | 981 (ND–3014) |
| **K** | 1431 (191–3833) | 500 (ND –1967) | 600 (ND–1682) |
| **Ca** | 1164 (ND –3204) | 397 (ND –1671) | 666 (ND–2160) |
| **Cr** | 23 (1–60) | 13 (1–60) | 20 (ND –54) |
| **Mn** | 30 (ND –64) | 17 (ND –49) | 33 (4–175) |
| **Fe** | 962 (173–2056) | 581 (140–1408) | 1269 (240–3578) |
| **Co** | 0.45 (0.03–1.06) | 0.23 (0.01–0.78) | 0.59 (0.07–1.74) |
| **Ni** | 7.3 (2.3–14.8) | 4.6 (ND –16.1) | 6.6 (ND –25.9) |
| **Cu** | 181 (7–390) | 109 (7–308) | 188 (32 –976) |
| **Zn** | 284 (ND –673) | 110 (ND –279) | 193 (ND –716) |
| **As** | 2.8 (0.06–5.7) | 1.9 (ND –7.1) | 2.2 (ND –7.9) |
| **Se** | 5.6 (ND –13.2) | 2.6 (ND –7.5) | 2.6 (ND –7.9) |
| **Rb** | 5.7 (0.4–12.3) | 2.2 (0.1–8.9) | 2.6 (0.2–8.9) |
| **Sr** | 6.6 (0.4–13.4) | 3.0 (0.2–12.2) | 4.8 (0.4–14.3) |
| **Cd** | 2.5 (0.2–15.1) | 0.8 (0.1–3.0) | 1.2 (0.2–10.6) |
| **Sn** | 19.5 (3.2–40.2) | 8.8 (0.3–35.9) | 12.3 (1.6–41.8) |
| **Cs** | 0.28 (0.07–1.01) | 0.14 (ND –0.51) | 0.19 (0.02–0.77) |
| **Tl** | 0.21 (ND –0.75) | 0.13 (ND –0.38) | 0.15 (0.03 –0.65) |
| **Pb** | 54 (3–172) | 31 (3–71) | 42 (4–176) |
| **Bi** | 0.76 (0.06–3.03) | 0.47 (ND–3.03) | 0.83 (0.12–3.24) |

*<DL - below detection limit. ND - not detected.*









**Table 5**. Concentrations of PAHs and derivatives for all campaigns.

| (ng m$^{-3}$) | Int$_{2.5}$ Average (Min - Max) | Ext$_{2.5}$ Average (Min - Max) | Ext$_{10}$ Average (Min - Max) |
|---|---|---|---|
| Nap | 0.30 (0.17–0.77) | 0.36 (0.02–0.77) | 0.41 (0.09–0.81) |
| Acy | 0.09 (0.06–0.12) | 0.10 (0.03–0.19) | 0.12 (0.05–0.34) |
| Ace | 0.03 (0.02–0.08) | 0.05 (0.02–0.16) | 0.07 (0.02-0.23) |
| Flu | 0.27 (0.15–1.03) | 0.31 (0.06–1.44) | 0.51 (0.10–1.75) |
| Phe | 0.65 (0.30–2.48) | 0.74 (0.12–3.55) | 1.28 (0.28–4.08) |
| Ant | 0.17 (0.10–0.44) | 0.16 (0.06–0.60) | 0.25 (0.08–0.67) |
| Flt | 0.48 (0.21–0.86) | 0.53 (0.06–1.40) | 0.73 (0.19–2.21) |
| Pyr | 0.52 (0.20–0.99) | 0.54 (0.07–1.54) | 0.71 (0.19–2.45) |
| BaA | 1.0 (0.3–2.4) | 0.9 (0.1–4.8) | 1.2 (0.3–5.9) |
| Chr | 1.8 (0.5–4.4) | 1.6 (0.3–5.7) | 2.1 (0.5–10.5) |
| BbF | 3.0 (0.9–6.1) | 2.3 (0.5–6.4) | 3.0 (0.7–13.3) |
| BkF | 2.5 (0.6–5.2) | 1.9 (0.2–7.4) | 2.5 (0.4–11.8) |
| BeP | 2.8 (0.6–6.1) | 2.2 (0.3–7.3) | 2.8 (0.5–14.4) |
| BaP | 2.3 (0.4–5.5) | 1.6 (0.2–7.6) | 2.0 (0.3–12.5) |
| Per | 0.35 (0.04–0.79) | 0.27 (<DL–1.27) | 0.38 (0.05–1.90) |
| InP | 2.9 (0.6–6.0) | 1.8 (0.3–6.3) | 2.4 (0.4–13.2) |
| DBA | 0.8 (0.1–2.3) | 0.6 (0.0–2.0) | 0.9 (0.0–5.1) |
| BPe | 2.4 (0.5–4.8) | 1.6 (0.2–5.5) | 2.1 (0.4–10.5) |
| Cor | 1.0 (0.1–2.4) | 0.7 (0.0–2.4) | 0.9 (0.1–5.2) |
| Total | 23.3 (6.0–48.8) | 18.4 (2.6–61.6) | 24.3 (5.4–115.3) |
| BaPE | 3.4 (0.6–8.0) | 2.4 (0.3–10.5) | 3.2 (0.5–18.3) |
| | | | |
| 1-NNap | <DL | <DL | <DL |
| 1-Methyl-4-NNap | <DL | <DL | <DL |
| 2-NNap | <DL | <DL | <DL |
| 2-NBP | 0.56 (<DL–1.36) | 0.56 (<DL–1.36) | 1.23 (0.47–2.47) |
| 1-Methyl-5-NNap | 0.18 (<DL–0.28) | <DL | <DL |
| 1-Methyl-6-NNap | 0.36 (<DL–0.40) | 0.27 (<DL–0.41) | 0.29 (<DL–0.86) |
| 2-Methyl-4-NNap | 0.45 (<DL–0.45) | 0.36 (<DL–0.44) | 0.42 (<DL–1.26) |
| 3-NBP | 0.60 (0.48–0.88) | 0.52 (<DL–0.87) | 0.55 (<DL–1.58) |
| 4-NBP | <DL | <DL | 0.18 (<DL–0.41) |
| 5-NAce | <DL | <DL | 0.20 (<DL–0.52) |
| 2-NFlu | 0.98 (0.78–1.39) | 0.99 (0.38–1.56) | 1.09 (0.54–1.79) |
| 2-NPhe | 0.43 (0.30–0.67) | 0.51 (0.19–1.40) | 0.61 (<DL–1.80) |
| 3-NPhe | 0.43 (<DL–0.46) | 0.44 (<DL–0.68) | 0.47 (<DL–1.11) |
| 9-NPhe | 0.62 (<DL–0.64) | 0.52 (<DL–0.64) | 0.55 (<DL–0.82) |
| 2-Nant | 0.66 (<DL–0.80) | 0.56 (<DL–0.80) | 0.61 (<DL–0.88) |
| 9-Nant | 0.44 (<DL–0.57) | 0.42 (<DL–0.69) | 0.46 (<DL–1.15) |
| 2-NFlt | 1.19 (<DL–1.35) | 0.98 (<DL–1.25) | 1.02 (<DL–1.43) |
| 3-NFlt | 1.45 (<DL–1.48) | 1.05 (<DL–1.48) | 1.02 (<DL–1.11) |
| 1-NPyr | 0.98 (<DL–1.12) | 0.73 (<DL–0.88) | 0.79 (<DL–1.28) |





| | | | |
|---|---|---|---|
| **2-NPyr** | 0.94 (<DL–0.99) | 0.76 (<DL–0.99) | 0.78 (<DL–1.27) |
| **4-NPyr** | 1.61 (<DL–1.67) | 1.27 (<DL–1.34) | 1.28 (<DL–1.72) |
| **7-NBaA** | 1.19 (<DL–1.34) | 0.91 (<DL–1.06) | 1.01 (<DL–1.67) |
| **6-NChr** | <DL | 0.60 (<DL–0.67) | 0.69 (0.58–1.10) |
| **3-NBA** | <DL | <DL | <DL |
| **6-NBaPyr** | <DL | <DL | 1.01 (<DL–1.19) |
| **1-NBaPyr** | <DL | <DL | <DL |
| **3NBePyr** | <DL | <DL | <DL |
| | | | |
| **1,4-BQ** | <DL | <DL | <DL |
| **1,4-NQ** | 0.54 (0.43–0.72) | 0.44 (0.28–0.67) | 0.46 (0.31–1.08) |
| **1,2-NQ** | <DL | <DL | <DL |
| **9,10-AQ** | 1.6 (0.8–3.7) | 2.5 (0.3–8.0) | 2.6 (0.4–10.9) |
| **9,10-PQ** | <DL | <DL | <DL |
| | | | |
| **Total PAHs/OC (%)** | 0.23 | 0.27 | 0.31 |
| **ΣLMW/ΣHMW** | 0.32 | 0.41 | 0.43 |
| **Flt/(Flt+Pyr)** | 0.5 | 0.5 | 0.5 |
| **BaA/Chr** | 0.5 | 0.6 | 0.5 |
| **InP/(InP+BPe)** | 0.5 | 0.5 | 0.5 |
| **BaP/(BaP+BeP)** | 0.4 | 0.4 | 0.4 |
| **BPe/BaP** | 1 | 1 | 1 |
| **2-NFlt /1-NPyr** | 1.3 | 1.3 | 1.3 |



