# Peer review of "Particulate pollutants in the Brazilian city of São Paulo: One-year investigation for the chemical composition and source apportionment"

_Atmospheric Chemistry and Physics, 2017_

## Referee Comment (RC1) · Anonymous Referee #1 · 19 May 2017

General comment

This paper reports an analysis of PM2.5 and PM10 chemical compositions in Sao Paulo (Brazil). The information is used for source apportionment using PMF receptor model and to investigate some atmospheric processes. The approach is not particularly new, however, the chemical investigation is very detailed and used to interpret some processes involving fine and coarse fractions of aerosol. Therefore, I believe that it would be interesting for readers and scientist. There are some parts of the paper that are not completely clear and interpretations not fully discussed (see my specific comments).

[Figure]

In conclusion, I suggest to consider the paper for publication only after a revision that addresses all my specific comments.

Specific comments

1) Number of chapters and subchapters are inconsistent.

2) It would be better to mention that also metals are analysed in the abstract otherwise the reader should wait several pages before to understand that also some metals are investigated.

3) Section 1.2. The height of the back-trajectories used start at 500 m that are not exactly near the ground. Why a lower starting point has not been chosen?

4) Section 1.4 (line 180). It is reported that PMF was applied to PM10 but I understand that all data including PM2.5 were used together. Please correct this incongruence.

5) Equation (3). The symbol EC was already used for elemental carbon, another symbol should be used in this equation otherwise it is confusing.

6) Section 3.1 line 210. Why correlation with minimum relative humidity and not with the average relative humidity measured during the collection of samples?

7) Section 3.1 lines 226-227. It is not clear why a comparison with London and Madrid? If a comparison with other urban areas is needed it would be better to expand it or explain why choosing specific towns.

8) Section 3.3 lines 275-279. This aspect should be discussed in more detail, are authors suggesting that the missing negative charges could be associated to carbonates, like calcium and/or magnesium carbonates? This could be put in evidence looking at the correlation between anion deficit and nss-Ca2+ concentrations as done, for example, in Contini et al. (Science of the Total Environment 472, 2014, pp. 248–261).

9) Section 3.3. line 312. What are IC2.5 and EC2.5?
10) Section 3.3. Line 298-299. This sentence is strange. It is not clear if authors speaks about PM2.5 or PM10.

11) Section 3.3. Line 323. I do not believe that there are anomalous enrichments, there are only enriched or not enriched elements if a single threshold approach is used.

12) Section 3.4. Lines 342-343. The ratio OC/EC is also strongly depending on the measurement protocol used. This aspect should be discussed.

13) Section 3.3. Lines 345-349. The mass closure analysis has been done considering metal oxides and possible contribution of carbonates like done, for example, in Cesari et al. (Environmental Science and Pollution Research 23, 15, pp 15133–15148, 2016)? The Si concentrations could be evaluated starting from Al concentrations. I believe that some more details are necessary to actually interpret the unaccounted mass.

14) Section 3.4.3 regarding the PAHs risks. It could be useful if the calculated values are compared with typical values found in other areas, see for example, Gregoris et al (Science of the Total Environment 476–477 (2014) 393–405) and references therein.

15) Section 3.6 Lines 471-475. It is not clear why only a few species are included in this analysis and not all measured components. This is a little confusing, I can understand that some species were eliminated in PMF for one reason or the other but it is not clear why the other Figures and Tables reports different groups of species.

16) Section 3.7 line 503. Why it is necessary to add such a large uncertainty? Could this be due to an underestimation of the uncertainties of chemical determination of the different species?

17) Section 3.7 lines 515-516. Mention what is the relative change in Q due to the application of constraints.

18) Section 4 line 567. Better high than higher.

19) Table 2 is not very useful because the detection limits vary for several order of magnitudes and it is not clear which species have low detection limits and which species have high detection limits. I believe that a more descriptive table could be constructed.

20) Table 4. It is not clear the difference between under detection limit and not detected. Essentially not detected for what reason? Because it is under detection limit?

21) The correlation table in supplementary materials is not readable too many small numbers.

22) In table S7. I do not understand the last line with the percentages.

---

## Referee Comment (RC2) · Anonymous Referee #2 · 15 Jun 2017

General comments

The manuscript entitled "Airborne particles in the Brazilian city of São Paulo: One-year investigation for the chemical composition and source apportionment" by Pereira et al. investigates the chemical composition of PM2.5 and PM10 filter samples collected in an extensive and an intensive campaign on a rooftop inside the USP campus in Sao Paulo City over one year. This year was characterized by low precipitation, high temperatures during the summer, resulting in high concentrations of air pollutants over the whole year. Source apportionment of a high number of investigated pollutants

was performed using PMF, where 5 factors could be identified. Findings were also used for health risk assessments. Although the number of samples is limited over the year, the results give a good and comprehensive general overview of the amount and sources of a high number of investigated pollutants in Sao Paulo City over the seasons. Overall, the work is within the scope of work published by ACP. Therefore, I recommend publication once the comments and questions below are addressed.

The manuscript will need copy-editing, as there are numerous grammatical errors, a few are mentioned below. In addition, some of the figures and tables are not readable without zooming in and should be revised.

Specific comments:

1- Revise the numeration of chapters and subchapters.

2- Abstract: Please explain in short the location of the sampling site in the abstract, and that there is only one site where the samples were collected. As Sao Paulo is a megacity with plenty of possible sites with potentially different sources of pollutants, it should be clear for the reader, that this study is different from several other papers publishing e.g. measurements in tunnels in Sao Paulo. The word airborne in the title and abstract is a bit misleading, as it is mostly associated with measurements on airplanes. It is very important to give the sampling height in the abstract and chapter 1.1.

3- Page 2, line 67: please write which primary sources PAHs have.

4- Table 2: As some species measured from the same instrument have apparently huge DL-differences, it is not helpful to just give the range of DLs. Additionally, add horizontal lines between the different instruments for a better separation.

5- Table 2: For the IC, do all ions really have exactly the same DL (6.4)? If so, the authors should write (in short) the reason for equal DLs and/or refer to a publication for this measurement technique.

6- Page 3, line 135: I guess, all carbon should be 'oxidized' to $CO_2$, not 'volatilized'?

7- Page 5, line 188: At that point, it is not clear for me how "missing data" occur. Do you mean missing filter samples for certain days, or missing data from specific species of a filter sample? If it is the last, how can that happen? Were there some species from a filter measured, but others not?

8- Table S1: Please explain "Therm. Ampl." and the possible reason for its correlation to PM2.5 and PM10. I am asking as these values are highlighted and not mentioned anywhere in the manuscript.

9- Page 6, line 229: Here the authors should make clear, that their data set does not have the data coverage over the year to be fully representative and comparable with the mentioned guidelines. To my knowledge, for both the EU and WHO guidelines, a coverage of 75% for one day and 90% for one calendar year, respectively, is mandatory for a proper risk assessment. Regarding the annual mean limits, the presented data is quite far away from that, considering there is only one day of sampling per week during the extensive campaign.

10- Page 7, line 279: Which other cationic species are you suggesting?

11- Page 8, line 291: Where and at which height were of gaseous species from CETESB measured? Collocated to the filter sampling?

12- Page 8, line 298: A R=0.35 is not relatively higher than R=0.78. Which correlation(s) are you comparing with R=0.35?

13- Table 4: What is the difference between "below detection limit" and "not detected?" Beside that "DL" is not used in table 4, but solely in Table 5, and there, "DL" is not explained.

14 - Page 9, line 347: If the unaccounted part may be attributed to absorbed water, can you see a correlation between this part and ambient humidity?

15 - Page 10, line 447: I do not think it is necessary to mention the location of a chamber (Florida), but rather the institute.

16- Page 13, line 500ff: The authors mention the strong and weak variables. Do all species, which are not mentioned, belong to the bad variables or are they simple not taken into account? Why and how did you define the 25% additional uncertainty? You combine species from different instruments, which have different uncertainties, as you pointed out in Table 2. Inserting these uncertainties into the error matrix usually cause that PMF strongly under- and overestimated the importance of variables within the solutions. Did you scale in some way the uncertainties to each other? Please explain also, why the organic character of Lev and Man justify to set them as weak variables.

17- Table S7: What do the percentages mean in the last row?

18- General aspects to PMF: I understood that the source of both the RD and VE factor is the traffic. It is not clearly seen in the polar plots (maybe because of the very high RD concentration point coming from NW) how well both factors are correlated to each other. The authors should add a comment on a possible correlation. I also do not understand why all factors were not compared to the gaseous CO and NOx time series, as these gases are commonly used to be correlated with primarily emitted factors, especially traffic sources. This would also justify additionally the authors choice of the 5-factor-solution.

Technical corrections:

Page 2, line 63. Revise this sentence: "Particulate organic carbon includes key species including polycyclic aromatic hydrocarbons (PAHs) and monosaccharides. The last are considered as biomass burning tracers (such as levoglucosan, mannosan, and galactosan)".

Page 3, line 118: use the present tense: intensifies

Page 3, line 118: Revise this sentence: "is presented" should be in the end of the sentence.

Table 1: Its hard to distinguish and attribute some information of the columns to the

sources. Add horizontal lines between the rows belonging to each source.

Page 6, line 208. Likewise above: Revise this sentence: "is presented" should be in the end of the sentence.

Figure 1: The 1-2 highest precipitation values increase the scale of the plot too large to see any more variations. Revise it to make the plot more readable, e.g. by changing the y-axis scale above 10 mm.

Page 6, line 234: between the words "study" and "was", the word "it" is missing.

Page 7, line 272: Revise the sentence beginning with "It has..."

Page 7, line 276: Begin the sentence with "The ratio..."

Page 7, line 283: write "potassium ions"

Page 8, line 288: write "the intensive campaign"

Table S2: This table is not readable as the letters and values are too small.

Page 8, line 305: write "increase of..."

Page 9, line 338: delete the word "be"

Figure 9a: Write the factors in the legend on the top of the graph in opposite order, in the figure caption as well, because RD is factor 1, IN is factor 2 etc.

Page 13, line 534: Revise the first part of the sentence.

Page 13, line 542: I guess there is the word "but" missing between the comma and "more"

Page 13, line 549: Rephrase the part in brackets into something like "(as seen by high mass loadings of...)"

Figure S3: Neither the axes nor the legends are readable.

---

## Author Response (AR1)

**(1) comments from Referees**

**Anonymous Referee #1**

**General comment**

This paper reports an analysis of $PM_{2.5}$ and $PM_{10}$ chemical compositions in Sao Paulo (Brazil). The information is used for source apportionment using PMF receptor model and to investigate some atmospheric processes. The approach is not particularly new, however, the chemical investigation is very detailed and used to interpret some processes involving fine and coarse fractions of aerosol. Therefore, I believe that it would be interesting for readers and scientist. There are some parts of the paper that are not completely clear and interpretations not fully discussed (see my specific comments). In conclusion, I suggest to consider the paper for publication only after a revision that addresses all my specific comments.

**Specific comments**

1- Number of chapters and subchapters are inconsistent.

2- It would be better to mention that also metals are analysed in the abstract otherwise the reader should wait several pages before to understand that also some metals are investigated.

3- Section 1.2. The height of the back-trajectories used start at 500 m that are not exactly near the ground. Why a lower starting point has not been chosen?

4- Section 1.4 (line 180). It is reported that PMF was applied to $PM_{10}$ but I understand that all data including $PM_{2.5}$ were used together. Please correct this incongruence.

5- Equation (3). The symbol EC was already used for elemental carbon, another symbol should be used in this equation otherwise it is confusing.

6- Section 3.1 line 210. Why correlation with minimum relative humidity and not with the average relative humidity measured during the collection of samples?

7- Section 3.1 lines 226-227. It is not clear why a comparison with London and Madrid? If a comparison with other urban areas is needed it would be better to expand it or explain why choosing specific towns.

8- Section 3.3 lines 275-279. This aspect should be discussed in more detail, are authors suggesting that the missing negative charges could be associated to carbonates, like calcium and/or magnesium carbonates? This could be put in evidence looking at the correlation between anion deficit and nss-$Ca^{2+}$ concentrations as done, for example, in Contini et al. (Science of the Total Environment 472, 2014, pp. 248–261).

9- Section 3.3. line 312. What are $IC_{2.5}$ and $EC_{2.5}$?

10- Section 3.3. Line 298-299. This sentence is strange. It is not clear if authors speaks about $PM_{2.5}$ or $PM_{10}$.

11- Section 3.3. Line 323. I do not believe that there are anomalous enrichments, there are only enriched or not enriched elements if a single threshold approach is used.

12- Section 3.4. Lines 342-343. The ratio OC/EC is also strongly depending on the measurement protocol used. This aspect should be discussed.

13- Section 3.3. Lines 345-349. The mass closure analysis has been done considering metal oxides and possible contribution of carbonates like done, for example, in Cesari et al. (Environmental Science and Pollution Research 23, 15, pp 15133–15148, 2016)? The Si concentrations could be evaluated starting from Al concentrations. I believe that some more details are necessary to actually interpret the unaccounted mass.

14- Section 3.4.3 regarding the PAHs risks. It could be useful if the calculated values are compared with typical values found in other areas, see for example, Gregoris et al (Science of the Total Environment 476–477 (2014) 393–405) and references therein.

15- Section 3.6 Lines 471-475. It is not clear why only a few species are included in this analysis and not all measured components. This is a little confusing, I can understand that some species were eliminated in PMF for one reason or the other but it is not clear why the other Figures and Tables reports different groups of species.

16- Section 3.7 line 503. Why it is necessary to add such a large uncertainty? Could this be due to an underestimation of the uncertainties of chemical determination of the different species?

17- Section 3.7 lines 515-516. Mention what is the relative change in Q due to the application of constraints.

18- Section 4 line 567. Better high than higher.

19- Table 2 is not very useful because the detection limits vary for several order of magnitudes and it is not clear which species have low detection limits and which species have high detection limits. I believe that a more descriptive table could be constructed.

20- Table 4. It is not clear the difference between under detection limit and not detected. Essentially not detected for what reason? Because it is under detection limit?

21- The correlation table in supplementary materials is not readable too many small numbers.

22- In table S7. I do not understand the last line with the percentages

**Anonymous Referee #2**

##

**General comments**

The manuscript entitled "Airborne particles in the Brazilian city of São Paulo: One-year investigation for the chemical composition and source apportionment" by Pereira et al. investigates the chemical composition of $PM_{2.5}$ and $PM_{10}$ filter samples collected in an extensive and an intensive campaign on a rooftop inside the USP campus in Sao Paulo City over one year. This year was characterized by low precipitation, high temperatures during the summer, resulting in high concentrations of air pollutants over the whole year. Source apportionment of a high number of investigated pollutants was performed using PMF, where 5 factors could be identified. Findings were also used for health risk assessments. Although the number of samples is limited over the year, the results give a good and comprehensive general overview of the amount and sources of a high number of investigated pollutants in Sao Paulo City over the seasons. Overall, the work is within the scope of work published by ACP. Therefore, I recommend publication once the comments and questions below are addressed.

The manuscript will need copy-editing, as there are numerous grammatical errors, a few are mentioned below. In addition, some of the figures and tables are not readable without zooming in and should be revised.

**Specific comments:**

1- Revise the numeration of chapters and subchapters.

2- Abstract: Please explain in short the location of the sampling site in the abstract, and that there is only one site where the samples were collected. As Sao Paulo is a megacity with plenty of possible sites with potentially different sources of pollutants, it should be clear for the reader, that this study is different from several other papers publishing e.g. measurements in tunnels in Sao Paulo. The word airborne in the title and abstract is a bit misleading, as it is mostly associated with measurements on airplanes. It is very important to give the sampling height in the abstract and chapter 1.1.

3- Page 2, line 67: please write which primary sources PAHs have.

4- Table 2: As some species measured from the same instrument have apparently huge DL-differences, it is not helpful to just give the range of DLs. Additionally, add horizontal lines between the different instruments for a better separation.

5- Table 2: For the IC, do all ions really have exactly the same DL (6.4)? If so, the authors should write (in short) the reason for equal DLs and/or refer to a publication for this measurement technique.

6- Page 3, line 135: I guess, all carbon should be 'oxidized' to $CO_2$, not 'volatilized'?

7- Page 5, line 188: At that point, it is not clear for me how "missing data" occur. Do you mean missing filter samples for certain days, or missing data from specific species of a filter sample? If it is the last, how can that happen? Were there some species from a filter measured, but others not?

8- Table S1: Please explain "Therm. Ampl." and the possible reason for its correlation to PM2.5 and PM10. I am asking as these values are highlighted and not mentioned anywhere in the manuscript.

9- Page 6, line 229: Here the authors should make clear, that their data set does not have the data coverage over the year to be fully representative and comparable with the mentioned guidelines. To my knowledge, for both the EU and WHO guidelines, a coverage of 75% for one day and 90% for one calendar year, respectively, is mandatory for a proper risk assessment. Regarding the annual mean limits, the presented data is quite far away from that, considering there is only one day of sampling per week during the extensive campaign.

10- Page 7, line 279: Which other cationic species are you suggesting?

11- Page 8, line 291: Where and at which height were of gaseous species from CETESB measured? Collocated to the filter sampling?

**12-** Page 8, line 298: A R=0.35 is not relatively higher than R=0.78. Which correlation(s) are you comparing with R=0.35?

**13-** Table 4: What is the difference between "below detection limit" and "not detected?" Beside that "DL" is not used in table 4, but solely in Table 5, and there, "DL" is not explained.

**14-** Page 9, line 347: If the unaccounted part may be attributed to absorbed water, can you see a correlation between this part and ambient humidity?

**15-** Page 10, line 447: I do not think it is necessary to mention the location of a chamber (Florida), but rather the institute.

**16-** Page 13, line 500ff: The authors mention the strong and weak variables. Do all species, which are not mentioned, belong to the bad variables or are they simple not taken into account? Why and how did you define the 25% additional uncertainty? You combine species from different instruments, which have different uncertainties, as you pointed out in Table 2. Inserting these uncertainties into the error matrix usually cause that PMF strongly under- and overestimated the importance of variables within the solutions. Did you scale in some way the uncertainties to each other? Please explain also, why the organic character of Lev and Man justify to set them as weak variables.

**17-** Table S7: What do the percentages mean in the last row?

**18-** General aspects to PMF: I understood that the source of both the RD and VE factor is the traffic. It is not clearly seen in the polar plots (maybe because of the very high RD concentration point coming from NW) how well both factors are correlated to each other. The authors should add a comment on a possible correlation. I also do not understand why all factors were not compared to the gaseous CO and $NO_x$ time series, as these gases are commonly used to be correlated with primarily emitted factors, especially traffic sources. This would also justify additionally the author's choice of the 5-factor-solution.

**Technical corrections:**

Page 2, line 63. Revise this sentence: "Particulate organic carbon includes key species including polycyclic aromatic hydrocarbons (PAHs) and monosaccharides. The last are considered as biomass burning tracers (such as levoglucosan, mannosan, and galactosan)".

Page 3, line 118: use the present tense: intensifies

Page 3, line 118: Revise this sentence: "is presented" should be in the end of the sentence.

Table 1: Its hard to distinguish and attribute some information of the columns to the sources. Add horizontal lines between the rows belonging to each source.

Page 6, line 208. Likewise above: Revise this sentence: "is presented" should be in the end of the sentence.

Figure 1: The 1-2 highest precipitation values increase the scale of the plot too large to see any more variations. Revise it to make the plot more readable, e.g. by changing the y-axis scale above 10 mm.

Page 6, line 234: between the words "study" and "was", the word "it" is missing.

Page 7, line 272: Revise the sentence beginning with "It has..."

Page 7, line 276: Begin the sentence with "The ratio..."

Page 7, line 283: write "potassium ions"

Page 8, line 288: write "the intensive campaign"

Table S2: This table is not readable as the letters and values are too small.

Page 8, line 305: write "increase of..."

Page 9, line 338: delete the word "be"

Figure 9a: Write the factors in the legend on the top of the graph in opposite order, in the figure caption as well, because RD is factor 1, IN is factor 2 etc.

Page 13, line 534: Revise the first part of the sentence.

Page 13, line 542: I guess there is the word "but" missing between the comma and "more"

Page 13, line 549: Rephrase the part in brackets into something like "(as seen by high mass loadings of...)"

Figure S3: Neither the axes nor the legends are readable.

**(2) author's response**

**Author comments on referee #1**

1- Number of chapters and subchapters are inconsistent.

   **AC:** This was corrected for the reviewed version.

2- It would be better to mention that also metals are analyzed in the abstract otherwise the reader should wait several pages before to understand that also some metals are investigated.

   **AC:** Metals are mentioned on the reviewed abstract:

   "...thermal-optical analysis. Trace elements were determined by inductively coupled plasma mass spectrometry. The associated risks..."

3- Section 1.2. The height of the back-trajectories used start at 500 m that are not exactly near the ground. Why a lower starting point has not been chosen?

   **AC:** Using a lower starting point can be challenging, the height of trajectories was selected to avoid the characteristics of the urban surface (with high rugosity height inducing mechanical turbulence).

4- Section 1.4 (line 180). It is reported that PMF was applied to $PM_{10}$ but I understand that all data including $PM_{2.5}$ were used together. Please correct this incongruence.

   **AC:** The PMF was applied considering all data ($PM_{2.5}$ and $PM_{10}$), this incongruence was corrected in the reviewed text:

   "The widely used source apportionment model, positive matrix factorization (PMF), was applied to all samples dataset (Paatero and Tapper, 1994)."

5- Equation (3). The symbol EC was already used for elemental carbon, another symbol should be used in this equation otherwise it is confusing.

   **AC:** This symbol for Element Concentration was be changed from EC to C in the reviewed equation:

   "$Unc = ([EF \times C]^2 + [0.5 \times DL]^2)^{1/2}$. Where EF is the error fractions and C is the element concentration."

6- Section 3.1 line 210. Why correlation with minimum relative humidity and not with the average relative humidity measured during the collection of samples?

   **AC:** Correlations with average and minimum relative humidity will be included on the reviewed text:

   "There were moderate negative correlations between $PM_{10}$ and average wind speed, minimum and average relative humidity; and between $PM_{2.5}$ and average wind speed and minimum relative humidity (Table S1).

7- Section 3.1 lines 226-227. It is not clear why a comparison with London and Madrid? If a comparison with other urban areas is needed it would be better to expand it or explain why choosing specific towns.

**AC:** The authors tried to compare particulate matter concentrations with those found in metropolis in different continents, such as Europe and Asia. It will be rewritten in order to explain the choice over those cities:

"The average values for $PM_{2.5}$ were higher than those obtained in a year study done in traffic sites in two European metropolis: London and Madrid in 2005 (warm period: 19.40 and 20.63 $\mu g\ m^{-3}$ for $PM_{2.5}$, respectively) (Kassomenos et al., 2014). The European Union has a more restrictive control of pollutant emissions compared to Latin American countries, since an annual mean of 40 $\mu g\ m^{-3}$ is established for $PM_{10}$ and a limit value of 25 $\mu g\ m^{-3}$ is imposed for $PM_{2.5}$ (Kassomenos et al., 2014). However, these averages in São Paulo are lower than the observed in year-round studies performed in Chinese megacities…".

**8-** Section 3.3 lines 275-279. This aspect should be discussed in more detail, are authors suggesting that the missing negative charges could be associated to carbonates, like calcium and/or magnesium carbonates? This could be put in evidence looking at the correlation between anion deficit and $nss\text{-}Ca^{2+}$ concentrations as done, for example, in Contini et al. (Science of the Total Environment 472, 2014, pp. 248–261).

**AC:** Authors suggested that the missing charges are associated to calcium and magnesium carbonates. Unfortunately we were unable to determine ionic $Ca^{2+}$ and $Mg^{2+}$. The authors decided to withdraw this balance.

**9-** Section 3.3. line 312. What are $IC_{2.5}$ and $EC_{2.5}$?

**AC:** They were the previous terms for extensive and intensive campaigns, and were changed to $Int_{2.5}$, $Ext_{2.5}$ and $Ext_{10}$ for the final text. This was corrected for the reviewed version:

"As observed for $nss\text{-}K^{+}$, elemental K average concentration was more than twice higher in $Int_{2.5}$ than $Ext_{2.5}$ ($p < 0.05$)."

**10-** Section 3.3. Line 298-299. This sentence is strange. It is not clear if authors speaks about $PM_{2.5}$ or $PM_{10}$.

**AC:** This paragraph refers to $Ext_{2.5}$ (extensive fine, $PM_{2.5}$) and $Ext_{2.5\text{-}10}$ (extensive coarse, $PM_{2.5\text{-}10}$), and will be rewritten in order to make it clearer:

"$Na^{+}$ was strongly correlated with $Cl^{-}$ in $Ext_{2.5}$ ($R = 0.78$) and had relatively higher correlations with this species ($R = 0.35$) in $Ext_{2.5\text{-}10}$."

**11-** Section 3.3. Line 323. I do not believe that there are anomalous enrichments, there are only enriched or not enriched elements if a single threshold approach is used.

**AC:** This terminology was done similarly as in Pereira et al. (Atmospheric Environment 41, 2007, pp. 7837–7850) and is also found in Odabasi et al. (Atmospheric Environment 36, 2002, pp. 5841–5851).

**12-** Section 3.4. Lines 342-343. The ratio OC/EC is also strongly depending on the measurement protocol used. This aspect should be discussed.

**AC:** The OC/EC ratios were determined with equivalent measurement protocols in Pio et al. (Atmospheric Environment 45, 2011, pp. 6121–6132). We emphasize this in the reviewed version. Some improvements were done; the references Querol et al., 2013 and Viana et al. 2007 were replaced by Amato et al., 2016 (Atmospheric Chemistry and Physics 16, 2016, pp. 3289–3309), which is a work where equivalent techniques were used:

"Ratios lower than 1 are constantly observed in roadway tunnels and are assumed to describe the composition of fresh traffic emissions (Pio et al., 2011). Amato et al. 2016 found values ranging from 1.8 and 3.7 at the urban background sites using equivalent measurement protocols. It was attributed to the distance from main roads, which can increase the influence of secondary OC (Pio, 2011). In this way, the values for OC/EC found in the present study may be due to vehicle emissions with contribution of secondary organic aerosols."

**13-** Section 3.3. Lines 345-349. The mass closure analysis has been done considering metal oxides and possible contribution of carbonates like done, for example, in Cesari et al. (Environmental Science and Pollution Research

23, 15, pp 15133–15148, 2016)? The Si concentrations could be evaluated starting from Al concentrations. I believe that some more details are necessary to actually interpret the unaccounted mass.

**AC:** Only the oxides were considered for the mass closure analysis, as in Alves et al. (Atmospheric Research 153, 2015, pp. 134–144). The concentrations of Si were evaluated from Al considering a ratio of 3.4 as in Clements et al. (Atmospheric Environment 89, 2014, pp. 373–381), but it may have been overestimated this way; the total accounted mass exceeds in 20% the $PM_{2.5}$ concentration for the intensive campaign.

14- Section 3.4.3 regarding the PAHs risks. It could be useful if the calculated values are compared with typical values found in other areas, see for example, Gregoris et al (Science of the Total Environment 476–477 (2014) 393–405) and references therein.

**AC:** The authors improved the paragraph considering more references:

"DBA had the largest contribution to carcinogenic potential and BaP the highest to mutagenic potential. In studies performed in Italian urban areas, BaP was the compound that most contributed to total carcinogenicity in PM, although the TEF used for DBA was lower in those cases (Cincinelli et al., 2007; Gregoris et al., 2014). LCR from exposure to atmospheric PAH was estimated by multiplying BaP-TEQ and BaP-MEQ by the unit risk ($87\times10^{-6}$ (ng m$^{-3}$)$^{-1}$) for exposure to BaP established by WHO (de Oliveira Alves et al., 2015; WHO, 2000) (Fig. 5) and was possible to observe an increase during the intensive campaign. In all campaigns, the values observed were higher than the observed in studies done in the Amazon during dry season with events of biomass burning (de Oliveira Alves et al., 2015); studies done at different seasons in other urban areas as New York and Madrid pointed carcinogenic risks within the recommended by environmental and health agencies (Jung et al., 2010; Mirante et al., 2013)".

15- Section 3.6 Lines 471-475. It is not clear why only a few species are included in this analysis and not all measured components. This is a little confusing, I can understand that some species were eliminated in PMF for one reason or the other but it is not clear why the other Figures and Tables reports different groups of species.

**AC:** More than 80 species were determined in this study, so they had to be selected in order to be included in this graph because of the lack of space for it. The ratios were calculated for all species, but only the species more important were discussed on the text: for being source tracers, having higher concentrations and/or health effects.

16- Section 3.7 line 503. Why it is necessary to add such a large uncertainty? Could this be due to an underestimation of the uncertainties of chemical determination of the different species?

**AC:** The uncertainties were increased in order to avoid discarding measurements that have poor data quality, due to some measurements below detection limits, this procedure was done according to reference of Paatero and Hopke (Analytica Chimica Acta 490, 2003, pp. 277-289).

17- Section 3.7 lines 515-516. Mention what is the relative change in Q due to the application of constraints.

**AC:** The relative change in Q (dQ = 0.4 %) due to application of constraints is mentioned in the reviewed version:

"Lev and Man were pulled up maximally in the biomass burning factor in order to have a better separation between both factors, relative change in Q was of 0.4 %".

18- Section 4 line 567. Better high than higher.

**AC:** Corrected in the reviewed text.

19- Table 2 is not very useful because the detection limits vary for several order of magnitudes and it is not clear which species have low detection limits and which species have high detection limits. I believe that a more descriptive table could be constructed.

**AC:** The authors suggest keeping this table, showing the range of those detection limits. More than 80 species were determined.

20- Table 4. It is not clear the difference between under detection limit and not detected. Essentially not detected for what reason? Because it is under detection limit?

**AC:** The term "under detection limit" was adopted for all tables in the reviewed version.

21- The correlation table in supplementary materials is not readable too many small numbers.

**AC:** The table file in excel will be uploaded as supplementary material, so it can be better understood.

22- In table S7. I do not understand the last line with the percentages.

**AC:** These are the percentages over the reduction on the ratio Qrobust/Qexpected as the number of factor increases, in the Q value analysis. These percentages could be removed for the reviewed final version of supplementary information file.

**Author comments on referee #2**

1- Revise the numeration of chapters and subchapters.

**AC.** Corrected in the reviewed version.

2- Abstract: Please explain in short the location of the sampling site in the abstract, and that there is only one site where the samples were collected. As São Paulo is a megacity with plenty of possible sites with potentially different sources of pollutants, it should be clear for the reader, that this study is different from several other papers publishing e.g. measurements in tunnels in São Paulo. The word airborne in the title and abstract is a bit misleading, as it is mostly associated with measurements on airplanes. It is very important to give the sampling height in the abstract and chapter 1.1.

**AC.** Modifications in the text:

The title is changed to: "Particulate pollutants in the Brazilian city of São Paulo: One-year investigation for the chemical composition and source apportionment"

Part of the Abstract was modified to: "In order to evaluate the sources of particulate air pollution and related health risks, a year-round sampling was done at the University of São Paulo campus (20 m above ground level), a green area near an important expressway. The sampling was performed for $PM_{2.5}$ ($\leq 2.5$ μm) and $PM_{10}$ ($\leq 10$ μm) in 2014 through intensive (every day sampling in wintertime) and extensive campaigns (once a week for the whole year) with 24 h of sampling".

In the item 2 "Methodology", section 2.1. Sampling campaigns: "Aerosol samples were collected at a São Paulo site (SPA, 23°33′34″S and 46°44′01″W) located on the ˘ rooftop of the Atmospheric Sciences Department (about 20 m above ground level), at Institute of Astronomy and Atmospheric Sciences (IAG-USP) building, within the campus of University of São Paulo."

3- Page 2, line 67: please write which primary sources PAHs have.

**AC.** Modifications in the text:

"Particulate organic carbon includes key species including polycyclic aromatic hydrocarbons (PAHs) and monosaccharides. The last are considered as biomass burning tracers (such as levoglucosan, mannosan, and galactosan) (Simoneit et al., 1999). PAHs have natural sources (synthesis by plants and bacteria, degradation of

plants, forest fires and volcanic emissions), but are mostly emitted by anthropogenic sources in urban sites (such as domestic, mobile, industrial and agricultural sources) (Abdel-shafy and Mansour, 2016; Ravindra et al., 2008)."

**4-** Table 2: As some species measured from the same instrument have apparently huge DL-differences, it is not helpful to just give the range of DLs. Additionally, add horizontal lines between the different instruments for a better separation.

**AC.** More than 80 species were determined and it would be difficult to present all detection limits. That is why the authors presented in such way. Lines were added between the different instruments.

**5-** Table 2: For the IC, do all ions really have exactly the same DL (6.4)? If so, the authors should write (in short) the reason for equal DLs and/or refer to a publication for this measurement technique.

**AC.** I noticed there was a typo in this table, in fact the DL value is 1.27 ng m$^{-3}$ for all ions (corrected in the reviewed version), and this was calculated with basis on detection limit of 1 ng ml$^{-1}$ (per volume of analyte solution on the IC). These values were provided by the FMI research group and are based on the signal to noise ratio of the IC baseline. Similar DLs were observed for the ions Na$^+$, NH$_4^+$, K$^+$, Cl$^-$, NO$_3^-$, SO$_4^{2-}$ and MSA.

**6-** Page 3, line 135: I guess, all carbon should be 'oxidized' to CO$_2$, not 'volatilized'?

**AC.** Corrected in the reviewed version.

**7-** Page 5, line 188: At that point, it is not clear for me how "missing data" occur. Do you mean missing filter samples for certain days, or missing data from specific species of a filter sample? If it is the last, how can that happen? Were there some species from a filter measured, but others not?

**AC.** The group decided to determine elements after few samples (n = 8) were collected in fiber glass filters. In these filters it is not possible to determine trace elements (the blank filters presented high levels of elements). To avoid discarding these samples (other species) we decided to do the statistical treatment as suggested by Brown et al. (Science of Total Environment 518–519, 2015, pp. 626-635). This paper suggests that the missing values are replaced by the median values of the valid samples.

**8-** Table S1: Please explain "Therm. Ampl." and the possible reason for its correlation to PM$_{2.5}$ and PM$_{10}$. I am asking as these values are highlighted and not mentioned anywhere in the manuscript.

**AC.** This is the thermal amplitude on the sampling days (maximum temperature minus minimum temperature) and will be removed from the supplementary information.

**9-** Page 6, line 229: Here the authors should make clear, that their data set does not have the data coverage over the year to be fully representative and comparable with the mentioned guidelines. To my knowledge, for both the EU and WHO guidelines, a coverage of 75 % for one day and 90 % for one calendar year, respectively, is mandatory for a proper risk assessment. Regarding the annual mean limits, the presented data is quite far away from that, considering there is only one day of sampling per week during the extensive campaign.

**AC.** The comparisons were removed from the paragraph, but the WHO guidelines were kept cited as a reference. The text:

"In the extensive campaign, the PM mass concentrations exhibited a wide range of concentrations. The concentrations in the Ext$_{2.5}$ ranged from 8 to 78 µg m$^{-3}$ (average 30 µg m$^{-3}$), whereas Ext$_{10}$ values varied between 12 and 113 µg m$^{-3}$ (average 44 µg m$^{-3}$) (Fig. 3). The World Health Organization (WHO) recommends a daily limit for PM$_{10}$ of 50 µg m$^{-3}$ and of 25 µg m$^{-3}$ for PM$_{2.5}$, (WHO, 2006) while the Brazilian Environmental Agency (CONAMA) recommends a threshold of 150 µg m$^{-3}$ for PM$_{10}$ (CONAMA, 1990; Pacheco et al., 2017). When considering the CONAMA standards, only one day in the extensive campaign was near the target limit. The Ext$_{10}$ campaign was divided into two periods: dry (April to September) and rainy (October to March). It was observed that the average PM10 was 52 µg m$^{-3}$ in the dry period and of 35 µg m$^{-3}$ in the rainy period."

**10-** Page 7, line 279: Which other cationic species are you suggesting?

**AC.** The authors decided to remove the cation/anion ratio discussion from the manuscript. Important species are missing for a complete analysis.

**11-** Page 8, line 291: Where and at which height were of gaseous species from CETESB measured? Collocated to the filter sampling?

**AC.** The gaseous species data were taken from two monitoring stations (CETESB). The first one ($NO_x$) is located inside the university campus (800 m far from the sampling site, at ground level) and another station is 3.2 km far (for CO, at 2m height). The sources that affect these stations are similar to those that affect the sampling site (Marginal Pinheiros expressway and local traffic). Other studies identified high correlations of pollutants concentrations between IPEN station and IAG sampling site; Oyama et al, (Atmospheric Chemistry and Physics, 16, 2016, 14397-14408) and Vara-Vela et al. (Atmospheric Chemistry and Physics, 16, 2016, 777-797).

**12-** Page 8, line 298: A R=0.35 is not relatively higher than R=0.78. Which correlation(s) are you comparing with R=0.35?

**AC.** It was not properly written, leading to the misunderstanding that these correlations were compared, but they were not. This part was rewritten:

"$Na^+$ was strongly correlated with $Cl^-$ in $Ext_{2.5}$ (R = 0.78) and in $Ext_{2.5-10}$ presented a relatively moderate correlation (R = 0.35)".

**13-** Table 4: What is the difference between "below detection limit" and "not detected?" Beside that "DL" is not used in table 4, but solely in Table 5, and there, "DL" is not explained.

**AC.** Corrected in the reviewed version.

**14-** Page 9, line 347: If the unaccounted part may be attributed to absorbed water, can you see a correlation between this part and ambient humidity?

**AC.** This correlation was determined and it was very low (R=0.2).

**15-** Page 10, line 447: I do not think it is necessary to mention the location of a chamber (Florida), but rather the institute.

**AC.** Modifications in the text:

"The Lev/Man ratios are characteristic of each type of biomass. The ratios were similar to that obtained in a chamber study with sugarcane burning (Lev/Man = 10, Hall et al., 2012), and also to that reported for the 2013 intensive campaign (Lev/Man = 12, Pereira et al., 2017)."

**16-** Page 13, line 500: The authors mention the strong and weak variables. Do all species, which are not mentioned, belong to the bad variables or are they simple not taken into account? Why and how did you define the 25% additional uncertainty? You combine species from different instruments, which have different uncertainties, as you pointed out in Table 2. Inserting these uncertainties into the error matrix usually cause that PMF strongly under- and overestimated the importance of variables within the solutions. Did you scale in some way the uncertainties to each other? Please explain also, why the organic character of Lev and Man justify to set them as weak variables.

**AC.** Some species were rejected due to low S/N ratio and other could not be included due to the relative high number of species compared to the number of samples. Some variables were not considered in the analysis and considered as bad variables, because they are highly correlated with other variables (high co-linearity and redundancies) and don't give us more information related to the sources. PAHs could also not be included because of the great effect of gas-particulate partitioning that is influenced by meteorological conditions. Maybe it was not well understood, but the organic character of Lev and Man mean they are a little partitioned between gas and

particle and also can be decomposed in the atmosphere, as discussed in Pio et al. (Atmospheric Environment 42, 2008, pp. 7530-7543) that's why they were considered as weak species. The uncertainties were increased in order to avoid discarding measurements that have poor data quality, due to some measurements below detection limits; this procedure was done according to reference of Paatero and Hopke (Analytica Chimica Acta 490, 2003, pp. 277-289). The additional uncertainty was applied to all the variables increasing proportionally the uncertainties. Besides the increase of the uncertainty, the physical solution does not change, presenting the same factors and results. We concluded that the solution is stable, because the same sources could be identified in most of the solutions generated, with different additional uncertainties.

**17-** Table S7: What do the percentages mean in the last row?

**AC.** Percentage of reduction over Qrobust/Qexpected with the increase in the number of factors. It was withdraw in the new version (Supplementary information).

**18-** General aspects to PMF: I understood that the source of both the RD and VE factor is the traffic. It is not clearly seen in the polar plots (maybe because of the very high RD concentration point coming from NW) how well both factors are correlated to each other. The authors should add a comment on a possible correlation. I also do not understand why all factors were not compared to the gaseous CO and NOx time series, as these gases are commonly used to be correlated with primarily emitted factors, especially traffic sources. This would also justify additionally the author's choice of the 5-factor-solution.

**AC.** There was a weak correlation between RD and VE and maybe it should not be expected to have such correlation, since the aerosol emitted by RD has a larger aerodynamic diameter than the aerosol related to VE. RD sources are primary and VE can be primary but they also can be secondarily formed from vehicular primarily emitted precursors. RD can be influenced by traffic but also by high wind speed. High correlations were observed between CO and NOx with primary sources factors as VE and BB. It may be due to the difficulty in separating these factors from each other, since their contributions increase with meteorological conditions (wind direction). Correlations were moderate between the gases and IN and RD and no correlation was observed between these gases and the SP (secondary processes). These findings will be included in the PMF discussions (section 3.7):

"...Vehicular source seemed to increase with winds coming from the North and Northwest, passing by the expressway, but decreased with SE winds, as observed previously (Sánchez-Ccoyllo and Andrade, 2002). The polar plots profiles of VE and RD factors presented a different pattern, since the aerosol from road dust suspension has a larger aerodynamic diameter (Karanasiou et al., 2009) and tends to increase with wind speed".

"High correlations (R > 0.8) were observed between the gases CO and $NO_x$ and the primary sources factors VE and BB. These gases are related to vehicular emissions (Alonso et al., 2010) and the correlations with the biomass burning factors may be due to the fact that it increases with the same wind direction as the vehicular factor. No correlations were found between these gases and SP factor."

**(3) author's changes in manuscript**

The most relevant changes made in the text are listed bellow and can be better observed on the attached marked-up version of the revised manuscript:

1- References to "airborne particles" were removed (title and in line 22) as suggested by referee #2.
2- More information was added to the abstract (sampling location and analytical method for element determination) (lines 24-25 and 32-33).
3- The numeration of chapters and subchapters was revised.
4- More information was added to PAHs sources description in the introduction (section 1 / lines 65-67).
5- Sampling height is described in the reviewed methodology (section 2.1 / line 99).
6- The acronym for element concentration was changed from *EC* to *C* (section 2.4 / lines 198-199).
7- Correlations of PM with average humidity were included in the discussion (section 3.1 / lines 213-214).
8- Comparisons with WHO guidelines were removed, since referee #2 stated that the coverage of the samplings was not sufficient for a proper risk assessment (section 3.1 / lines 221-222 and 224-225).
9- This paragraph was modified in order to explain the choices over those sites (section 3.1 / lines 230-237). ∑cations/∑anions discussion was removed (section 3.3 / line 280-282). If available, $Ca^{2+}$ and $Mg^{2+}$ concentrations could explain the missing charges associated with carbonates.
10- OC/EC ratio discussion was revised (section 3.4 / lines 344-348). References from Querol et al., 2013 and Viana et al., 2007 were removed, Amato et al. 2016 was considered for the OC/EC ratios comparison.
11- PAHs risk assessment discussions were improved (section 3.5.3 / lines 441-443 and lines 446-449).
12- Relative change in Q value was added (section 3.8 / line 529).
13- A comment on polar plot profiles of VE and RD factors was included (section 3.8 / lines 552-554).
14- Discussions on the correlations between gases and PMF factors were inserted (section 3.8 / lines 563-568) A table was also inserted (Table S8).
15- The Y-axis of the precipitation plot (Figure 2c) was broken above 15 mm to make it more readable.
16- The legend was altered in order to correct the factors order (Figure 9a).
17- Horizontal lines were added between rows on the tables (Tables 1 and 2).
18- Detection limits for Ion Chromatography were corrected (Table 2).
19- The acronyms ND were changed to <DL (Table 4).
20- Table S2 will be submitted as an Excel file, since it was not readable in the Supplementary Information.
21- Figure S3 charts were enlarged in order to make them more legible in the Supplementary Information.
22- Other minor technical corrections were made along the text (as suggested by referees).

[revised manuscript text omitted]

---

## Author Response (AR2)

**(1) *Comments from referee 2:**

Re-Review (page and line numbers refer to the marked up version of the revised manuscript):

Overall, the authors have adequately addressed my comments and suggestions. I recommend the manuscript be published in ACP. However, there are just a few minor points that should be addressed before final publication:

to point 7: to justify this procedure, please add the reference Brown et al to the manuscript.

to point 9: The authors removed main parts of the comparison to EU and WHO guidelines. Although it sounds picky, please add a sentence with the required data coverage for each guideline in %, and where you state that your data set does not reach these coverages.

to point 11: This point was well explained by the authors. Please add the information you give here to the manuscript (if you prefer, in a shorter form) with the 2 references.

to point 16: Similar to point 11, please add your explanations in short form to the manuscript, which will also help the reader to understand the whole strategy of your PMF data pretreatments.

Technical corrections:

 Page 6, line 231: Use the plural of metropolis, there are several options.

Page 9, line 292: Between "higher" and "contribution" is a space too much.

Figure 9a: Write the factors in the figure caption in opposite order because RD is factor 1, IN is factor 2 etc.

**(2) *Author's response:**

The authors thank the referee for the suggestions.

Point 7 – This reference was added (line 193).

Point 9 – A reference about one calendar year coverage from European Environment Agency report was added in the paragraph (lines 223-225), although the authors were not able to find a reference regarding the coverage of 75% for one day.

Point 11 – Some discussions and references about the location and height of gaseous species monitoring were added to the paragraph (lines 296-299).

Point 16 – Those explanations about the PMF analysis were added in the discussions (lines 514-530).

The suggested technical corrections have been done in the reviewed version, among other corrections:

- Author affiliations.
- "Lev" and "Man" abbreviations were changed to the extended form in some paragraphs (lines 516 and 536).
- Minor changes (lines 62, 63, 217, 220, 233, 269, 270, 292, 343, 364, 369, 397, 418, 445, 452, 453, 457, 491, 499, 509, in table 2 and in figure 9a caption).

[revised manuscript text omitted]

---

## Author Response (AR3)

**Modifications in the text:**

**A change was done in the affiliation, as suggested by the author Prashant Kumar:**

[revised manuscript text omitted]